

# SMRT: An active / passive microwave radiative transfer model for snow with multiple microstructure and scattering formulations (v1.0)

Ghislain Picard[1], Melody Sandells[2], and Henning Löwe[3]

[1]UGA, CNRS, Institut des Géosciences de l'Environnement (IGE), UMR 5001, Grenoble, F-38041, France
[2]CORES Science and Engineering Limited, Burnopfield, UK
[3]WSL Institute for Snow and Avalanche Research SLF, Davos, Switzerland

*Correspondence to:* Ghislain Picard (ghislain.picard@univ-grenoble-alpes.fr)

**Abstract.** The Snow Microwave Radiative Transfer (SMRT) thermal emission and backscatter model was developed to determine uncertainties in forward modeling through intercomparison of different model ingredients. The model differs from established models by the high degree of flexibility in switching between different electromagnetic theories, representations of snow microstructure, and other modules involved in various calculation steps. SMRT v1.0 includes the Dense Media Radiative

Transfer theory (DMRT), the Improved Born Approximation (IBA) and independent Rayleigh scatterers to compute the intrinsic electromagnetic properties of a snow layer. In the case of IBA, five different formulations of the autocorrelation function to describe the snow microstructure characteristics are available, including the sticky hard sphere model, for which close equivalence between IBA and DMRT theories has been shown here. Validation is demonstrated against established theories and models. SMRT was used to identify that several former studies conducting simulations with in-situ measured snow properties are now comparable and moreover appear to be quantitatively nearly equivalent. This study also proves that a third parameter

is needed in addition to density and specific surface area to characterize the microstructure. The paper provides a comprehensive description of the mathematical basis of SMRT and its numerical implementation in Python. Modularity supports model extensions foreseen in future versions comprising other media (e.g. sea-ice, frozen lakes), different scattering theories, rough surface models, or new microstructure models.

# 1   Introduction

The number and diversity of space-borne observations from passive and active microwave sensors over snow-covered regions has considerably increased over the last three decades. Due to the demand for global monitoring of the cryosphere and its change, numerous algorithms have been developed to retrieve geophysical information on snow cover extent (Grody and Basist, 1996; Nghiem and Tsai, 2001), snow depth and snow water equivalent on both land (Josberger and Mognard, 2002; Kelly and

Chang, 2003; Derksen et al., 2003) and sea ice (Comiso et al., 2003; Cavalieri et al., 2012), snow accumulation on ice sheets (Abdalati and Steffen, 1998; Vaughan et al., 1999; Drinkwater et al., 2001; Winebrenner et al., 2001; Flach et al., 2005; Arthern et al., 2006; Dierking et al., 2012) wet snow (Zwally, 1977; Shi and Dozier, 1995; Abdalati and Steffen, 1997; Nagler and



Rott, 2000; Steffen, 2004; Picard et al., 2007), snow temperature (Shuman et al., 1995; Schneider and Steig, 2002; Schneider et al., 2004), snow grain size (Brucker et al., 2010; Picard et al., 2012) and snow density (Schwank et al., 2015; Champollion et al., 2013). Even though many applications still rely on empirical approaches to relate snowpack properties (e.g. SWE) and measured signals, it is generally accepted that a physical understanding of the interaction of snow with electromagnetic waves

is necessary to improve the accuracy and overcome inherent difficulties of the retrieval as an underdetermined problem. The retrieval of snow properties is therefore often preceded by forward modeling and data assimilation (Durand and Margulis, 2007; Picard et al., 2009; Takala et al., 2011; Toure et al., 2011; Huang et al., 2012) to predict the satellite signal from prescribed snowpack properties that can be either obtained from measurements (e.g. Rosenfeld and Grody, 2000; Brucker et al., 2011a; Rees et al., 2010; Derksen et al., 2012, 2014; Kontu et al., 2014) or snow models (e.g. Flach et al., 2005; Brucker et al., 2011b;

Andreadis and Lettenmaier, 2012; Kang and Barros, 2012; Wójcik et al., 2008; Kontu et al., 2017). Since the influence of the atmosphere on the wave can be commonly neglected for most microwave frequencies, the actual modeling challenge lies in the snowpack and the underlying surface (soil, ice or water) where the coupling of various ingredients needs to be understood with sufficient accuracy to build efficient forward models. Examples comprise scattering by snow microstructure, liquid water, salinity, ice lenses (Montpetit et al., 2013), coherent effects (Mätzler, 1987; Leduc-Leballeur et al., 2015; Tan et al., 2015a),

the underlying surface and especially its roughness. All of these effects have to be taken into account by physically-based snow-microwave models.

Several physically-based models have been developed previously mainly for passive microwave remote sensing, including HUT (Lemmetyinen et al., 2010), MEMLS (Wiesmann and Mätzler, 1999), DMRT-QMS (Tsang et al., 2006; Liang et al., 2008), DMRT-ML (Picard et al., 2013) and other DMRT-based ones (Macelloni et al., 2001; Grody, 2008; Brogioni et al.,

2009). In addition, several models were tailored to low frequencies (i.e. up to a few GHz), such as 2S (Schwank et al., 2014), CMES (Drusch et al., 2009), WALOMIS (Leduc-Leballeur et al., 2015) and others (Tan et al., 2015a), triggered by the inception of space-borne L-band radiometry (Barre et al., 2008). Early models for active microwave observations include only single scattering mechanisms (Bingham and Drinkwater, 2000; Flach et al., 2005; Longepe et al., 2009; Lacroix et al., 2008) which is generally sufficient at low frequencies where scattering is weak compared to absorption. Only recently DMRT-QMS and

MEMLS have been adapted to an active mode that accounts for multiple scattering (Tsang et al., 2007; Liang et al., 2008; Proksch et al., 2015) which is particularly relevant for high-frequency radar such as SARAL AltiKa (Verron et al., 2015). The combined active/passive capability in the same model is particularly relevant for dual-mode missions such as SMAP (Entekhabi et al., 2010). The large number of different models is a natural consequence of both, the diversity of possible approaches at each stage of the calculation (e.g. effective snow permittivity, scattering, solution of the radiative transfer equation, etc.) and

the wide range of applications (e.g. research versus operational use). This results in a practical difficulty of choosing the most suitable model for a given application. In addition, the scope and comparability of predictions of the same property from different models must be taken with caution, given the differences in model ingredients.

As a remedy, more and more studies include predictions from different models (e.g. Wójcik et al., 2008; Rees et al., 2010; Roy et al., 2013; Kwon et al., 2015; Sandells et al., 2017), to draw more general conclusions. Other studies directly focused on

the inter-comparison of different models (Tedesco and Kim, 2006; Tse et al., 2007; Tian et al., 2010; Xiong and Shi, 2013; Pan





et al., 2016; Löwe and Picard, 2015; Sandells et al., 2017; Royer et al., 2017) to quantify the differences. Though insightful and necessary, these efforts did not lead to a reduction of the number of models as none of the studies considered the entirety of models and none showed a clear superiority of a single model. The latter fact was partly explained in Löwe and Picard (2015) who demonstrated the near-equivalence of two approaches, namely IBA (Mätzler, 1998) and DMRT (Tsang et al., 1985; Shih et al., 1997) that were previously considered to be different. This was achieved by relating the microstructural foundations of either approach, demonstrating the necessity to compare different microstructural formulations.

The representation of snow microstructure is critical since it immediately constrains the choice of formulation to compute the scattering coefficient. Several empirical formulations of the scattering coefficient have been developed as a function of traditional grain size (Hallikainen et al., 1987) or the exponential correlation length (Wiesmann et al., 1998). These formulations are available in HUT and MEMLS. But as for any empirical approach, the applicability is not guaranteed beyond the limits of calibration. This makes formulations based on fundamental principles (Maxwell equations) attractive. For instance, the Dense Media Radiative Transfer theory (DMRT; Tsang et al., 1985; West et al., 1993; Shih et al., 1997; Tsang et al., 2000a, 2007) is used by several models (e.g. DMRT-ML, DMRT-QMS, Longepe et al. (2009), etc.). DMRT represents snow as a collection of ice spheres whose relative positions is constrained by the sticky hard sphere (SHS) model. Thereby a stickiness parameter controls the propensity of the spheres to stick to each other and form clusters with higher scattering power that disperse grains. The stickiness has thus an impact on the validity of approximations when computing the scattering coefficient. Some DMRT-based models (e.g. DMRT-ML and Macelloni et al., 2001) are restricted to "short range" approximation which yields a close-form analytical solution for the scattering and absorption coefficients and the phase function. However, this approximation requires that both, grain (sphere) size and the cluster size are small compared to the wavelength. While this is reasonable for snow at frequencies below 19 GHz it is more problematic at higher frequencies (Grody, 2008). The "long range" approximation relaxes the constraint on cluster size which allows to treat highly sticky spheres at the cost of a numerical integration to compute the scattering coefficient and phase function. To our knowledge, this approximation is not implemented in any available model. To additionally relax constraints on grain size, the DMRT-QCA Mie formulation is needed (Tsang et al., 2000a), allowing simulations at frequencies higher than 37–89 GHz. DMRT-QMS is the only model to implement this advanced assumption. Despite the attractive features of the DMRT theory, the representation of snow microstructure by the SHS model has a major drawback. The stickiness parameter can not be easily retrieved from field measurements yet because microstructures of non-sticky spheres are not directly applicable to natural snow (Brucker et al., 2011b; Picard et al., 2014; Roy et al., 2013). Furthermore, estimating stickiness from high-resolution microstructure images – as obtained from X-ray micro-computed tomography ($\mu$CT) – appears to be numerically unstable (Löwe and Picard, 2015), leading to the conclusion that SHS is likely not a good representation for natural snow.

The Improved Born Approximation developed by Mätzler (1998) is an alternative approach to compute the scattering coefficient. It uses the same basic electromagnetic principles (Born approximation) as DMRT but it is not limited to a particular microstructure model. Instead of employing a particle model and characterizing their relative positions through the pair-correlation function as in DMRT, IBA uses the relative position of the ice material directly, which is mathematically captured by the autocorrelation function (ACF) of the ice indicator function (Torquato and Haslach, 2002; Löwe and Picard, 2015).



In Mätzler (1998) the ACF of non-sticky overlapping spheres was investigated to obtain an analytical form for the scattering coefficient. However, in MEMLS (Mätzler and Wiesmann, 1999), the main model using IBA, the choice of ACF is limited to an exponential function that is characterized by a single parameter, the correlation length. The correlation length can be obtained from thin 2D sections of snow samples (Wang et al., 1998; Wiesmann et al., 1998) or µCT. Even though the measurements

are time-consuming, the estimation is numerically stable. On one hand, using only a single parameter to describe the whole microstructure seems advantageous over SHS which requires two parameters, size and stickiness. On the other hand, Mätzler (2002) had to propose different relationships between correlation length and surface-area-to-volume ratio to represent different snow types, demonstrating the ambiguity of the exponential correlation length and indicating the necessity of describing snow microstructure by at least two parameters. This is also reflected by more recent attempts that use level-cut Gaussian random

fields as microstructure model for a bicontinuous medium as an alternative to the SHS model (Ding et al., 2010; Chang et al., 2014, 2016). This approach is very flexible, but as for SHS, the link of model parameters to natural snow microstructure and in-situ measurement techniques remains to be understood (Chang et al., 2016). This requires a comparison of different microstructure models in the context of a chosen scattering theory. Due to the near-equivalence of IBA and DMRT (Löwe and Picard, 2015) it seems reasonable then to utilize IBA together with a library of ACFs as candidates to represent natural snow.

All examples mentioned above indicate a clear demand for a modular and extensible approach that unifies existing knowledge and facilitates efficient intercomparisons of model ingredients with particular focus on the representation of microstructure. To this end we developed the Snow Microwave Radiative Transfer (SMRT) model, as a versatile tool to compute backscattering and brightness temperature (active/passive mode) from multi-layered media, composed of bicontinuous, random microstructures (typically snow or bubbly ice), overlying a reflective surface (typically soil, water or ice). The originality of this new

model is the flexibility for the user to select between various electromagnetic or microstructure formulations at different stages of the forward modeling problem. SMRT includes IBA, DMRT and independent Rayleigh scattering theories to compute the scattering and absorption coefficients and the phase function. When using IBA, it is possible to choose between several representations of isotropic microstructures that are prescribed by analytical forms of the ACF. This is complemented by several soil model implementations and permittivity formulations. Additionally, language bindings are implemented to facilitate a direct

comparison with widely-used models (DMRT-QMS, MEMLS and HUT) using their original code. In short, SMRT is designed to enable easy and rigorous inter-comparison and exploration of electromagnetic theories, common models and microstructure representations. SMRT version 1.0 is written in Python and released as open-source under the LGPLv3 license.

The paper is organized as follows. The next section gives an overview about the model architecture, the most important formulations, the code structure and basic usage. In the third section we present an inter-comparison of SMRT with other

models and explore the equivalence between different microstructures. The fourth section is dedicated to the discussion of limitations and perspectives. The last section concludes the paper.





## 2 SMRT description

SMRT was designed to be easy to use, computationally efficient and to allow exploration of the various approximations or formulations available for computing snow scattering and emission in the microwave domain. Even though the goal was to maximize flexibility and versatility, some specific choices and compromises were nevertheless necessary: i) SMRT is a radiative transfer model. This implies that inter-layer interferences and coherent effects are neglected. It is not suitable for interferometric computation. ii) SMRT considers media composed of plane-parallel, horizontally infinite, homogeneous layers and is therefore not suitable to compute 3D effects. iii) The current version is limited to isotropic media at the microstructure scale as well as at the scale of the snowpack. This means that microstructural anisotropy of snow is neglected (Leinss et al., 2016) and that structures formed by wind (sastrugies, dunes) are not taken into account yet. Even though SMRT is primarily designed for microwaves and snow, restrictions on spectral range and materials are nowhere made explicit to allow for future extensions to the optical range and other random media (sea-ice, layered soil, atmosphere). As a consequence of these decisions on design, the model is therefore composed of a fixed architecture, described in Section 2.1 and many switchable formulations described in Sections 2.2 and 2.3 and in Appendix 8.

### 2.1 Model architecture

The model is centered around the radiative transfer equation to compute the propagation of radiative energy in the medium produced by thermal emission in the medium (passive mode) and received from the sky (radar beam in active mode and sky thermal emission in passive mode). In addition to the radiative transfer equation, the other main components include the electromagnetic model that describes electromagnetic behavior of snow (i.e. the effective refractive index or permittivity, absorption and scattering coefficients and phase function), the boundary conditions between layers (called interfaces hereinafter) and at the bottom interface (called substrate hereinafter). All these components are well isolated in the code and various formulations from the literature are available. Here, only the common elements are presented, the switchable formulations are described in the following sections and appendix.

The model solves the time-independent radiative transfer equation assuming a horizontally homogeneous medium, this is:

$$\mu \frac{\partial \mathbf{I}(\mu, \phi, z)}{\partial z} = -\boldsymbol{\kappa}_{\mathrm{e}}(\mu, \phi, z)\,\mathbf{I}(\mu, \phi, z) + \frac{1}{4\pi} \iint\limits_{4\pi} \mathbf{P}(\mu, \phi; \mu', \phi', z)\,\mathbf{I}(\mu', \phi', z)\, d\Omega' + \boldsymbol{\kappa}_{\mathrm{a}}(\mu, \phi, z)\,\alpha T(z)\mathbf{1} \qquad (1)$$

where $\mathbf{I} = (I_{\mathrm{V}}, I_{\mathrm{H}}, U, V)$ is the specific intensity. $\mathbf{P}(\mu, \phi; \mu', \phi', z)$ is the 4×4 phase matrix. $\boldsymbol{\kappa}_{\mathrm{a}}$ and $\boldsymbol{\kappa}_{\mathrm{e}}$ are the absorption and extinction coefficients and the vector $\mathbf{1} = (1,1,1,1)$. The extinction coefficient is given by $\boldsymbol{\kappa}_{\mathrm{e}} = \boldsymbol{\kappa}_{\mathrm{s}} + \boldsymbol{\kappa}_{\mathrm{a}}$ where $\boldsymbol{\kappa}_{\mathrm{s}}$ is the scattering coefficient. Directions are defined by the cosine of the zenith angle $\mu$ and by the azimuthal angle $\phi$. The associated solid angle is $\Omega$. The $z$ axis is taken upward (as usual in Earth science), meaning that the incident beam and downwelling radiation have $\mu < 0$, while upwelling radiation has $\mu > 0$. This equation is valid in both active and passive modes in the microwave range. The brightness temperature $T_{\mathrm{B},p}$, with $p =$ H or V, is proportional to the intensity $I_p = \alpha T_{\mathrm{B},p}$ (Rayleigh-Jeans approximation) with $\alpha = 2\nu^2 k/c^2$ where $k$ and $c$ are the Planck constant and speed of light. In practice, for the passive





mode and by using the linearity of the equation (1), $I_p$ can be replaced by the brightness temperature and $\alpha$ set to 1. This is the case in our code.

Further assuming that i) snow is isotropic at the microscopic level ii) the medium is azimuthally symmetric and iii) the medium is composed of homogeneous layers (Fig. 1), the equation becomes:

$$\mu\frac{\partial \mathbf{I}^{(l)}(\mu,\phi,z)}{\partial z} = -\left(\boldsymbol{\kappa}_{\mathrm{s}}^{(l)}(\mu) + \boldsymbol{\kappa}_{\mathrm{a}}^{(l)}(\mu)\right)\mathbf{I}^{(l)}(\mu,\phi,z) + \frac{1}{4\pi}\iint\limits_{4\pi}\mathbf{P}^{(l)}(\mu,\mu',\phi-\phi')\mathbf{I}^{(l)}(\mu',\phi',z)\,d\Omega' + \boldsymbol{\kappa}_{\mathrm{a}}^{(l)}(\mu)T^{(l)}\mathbf{1}. \qquad (2)$$

Here $l = 1\ldots L$ denotes the layer index ranging from the top ($l = 1$) to the base ($l = L$).

The continuity conditions at layer interfaces and the boundary condition at the bottom interface are expressed by:

$$\mathbf{I}^{(l)}(\mu<0,\phi,z_{l-1}) = \mathbf{R}^{\mathrm{spec,top},(l)}(\mu)\mathbf{I}^{(l)}(-\mu,\phi,z_{l-1}) + \frac{1}{2\pi}\iint\limits_{2\pi,\mu'>0}\mathbf{R}^{\mathrm{diff,top},(l)}(\mu,\mu',\phi-\phi')\mathbf{I}^{(l)}(\mu',\phi',z_{l-1})\,d\Omega'$$

$$+\mathbf{T}^{\mathrm{spec,bottom},(l-1)}(\mu)\mathbf{I}^{(l-1)}(\mu,\phi,z_{l-1}) + \frac{1}{2\pi}\iint\limits_{2\pi,\mu'<0}\mathbf{T}^{\mathrm{diff,bottom},(l-1)}(\mu,\mu',\phi-\phi')\mathbf{I}^{(l-1)}(\mu',\phi',z_{l-1})\,d\Omega' \qquad (3)$$

$$\mathbf{I}^{(l)}(\mu>0,\phi,z_l) = \mathbf{R}^{\mathrm{spec,bottom},(l)}(\mu)\mathbf{I}^{(l)}(-\mu,\phi,z_l) + \frac{1}{2\pi}\iint\limits_{2\pi,\mu'<0}\mathbf{R}^{\mathrm{diff,bottom},(l)}(\mu,\mu',\phi-\phi')\mathbf{I}^{(l)}(\mu',\phi',z_l)\,d\Omega'$$

$$+\mathbf{T}^{\mathrm{spec,top},(l+1)}(\mu)\mathbf{I}^{(l+1)}(\mu,\phi,z_l) + \frac{1}{2\pi}\iint\limits_{2\pi,\mu'<0}\mathbf{T}^{\mathrm{diff,top},(l+1)}(\mu,\mu',\phi-\phi')\mathbf{I}^{(l+1)}(\mu',\phi',z_l)\,d\Omega' \qquad (4)$$

where $z_l$ is the $z$ position of the bottom of layer $l$ and conversely $z_{l-1}$ is the height of the top of the layer $l$. $\mathbf{R}$ and $\mathbf{T}$ are reflectivity and transmittivity matrix. The superscript "spec" denotes the specular (a.k.a. coherent) components and "diff" is the diffuse (a.k.a incoherent) components. For a perfectly flat interface, the diffuse component is zero and the specular component is given by the Fresnel coefficients (e.g. Jin, 1994, p. 59). The "top" superscript denotes the coefficients from a layer to the one above, and "bottom" denotes cofficients to the layer below.

Given the main governing equations (2), (3) and (4) it is instructive to summarize the architecture and main components of SMRT (Fig. 2). The quantities $\boldsymbol{\kappa}_{\mathrm{s}}$, $\boldsymbol{\kappa}_{\mathrm{a}}$ and $\mathbf{P}$ in the main equation (2) are computed independently for each layer by the *electromagnetic model* component (Fig. 2) using one of the implemented theories (in version 1.0 IBA, DMRT, Independent Rayleigh scattering) and input parameters characterizing the snow *microstructure* component. The inter-layer reflectivity and transmittivity coefficients in equations (3) and (4) are computed with the *interface* component (e.g. with Fresnel coefficients for flat interfaces) and with the *substrate* component. The effective refractive index needed for these calculations is given by the *electromagnetic model* component which in turn uses *materials permittivity* formulations of the raw materials (ice, water, air, ...). Once fully specified, the equations are numerically solved with the *radiative transfer equation solver* component which provides a numerical method adapted to the plane-parallel, multi-layer configuration, and the result, that is the intensity emerging in all or specific directions from the snowpack, is returned to the user. All formulations and methods for each component are described in the Appendix, except the Improved Born Approximation (one of the *electromagnetic model*s detailed in the next section) which is essential to understand the representation of snow microstructure in SMRT.





## 2.2 Improved Born Approximation

The implementation of the Improved Born Approximation (IBA) in SMRT closely follows the original work Mätzler (1998) with slight differences. The phase function in the 1-2 frame (Mätzler, 1998; Ding et al., 2010) is calculated for a 2-phase medium (subscript 1 denotes the host constituent and subscript 2 denotes the scattering constituent, e.g. air and ice are used for light snow) as:

$$p\,(\vartheta,\varphi)_{\text{1-2 frame}} = f_2(1-f_2)(\epsilon_2 - \epsilon_1)^2\, Y^2(\epsilon_1,\epsilon_2)\, k_0^4\, M(|\mathbf{k}_d|)\sin^2\chi \tag{5}$$

where the angles $(\vartheta, \varphi)$ denote the scattering direction if the incident direction is taken as polar axis. The free-space wavenumber is denoted by $k_0 = 2\pi\nu/c$ with the wave frequency $\nu$. The volume fraction of constituent 2 is denoted by $f_2$ and related to the medium density $\rho$ by $f_2 = \rho/\rho_2$. The relative permittivities of phases 1,2 are denoted by $\epsilon_1$ and $\epsilon_2$. The temperature and frequency dependence of the permittivity is taken into account but not made explicit in the notation. Polarization information is carried in the polarization angle $\chi$, which is the angle between the incident electric field and scattering direction. This angle is given by $\sin^2\chi = 1 - \sin^2\vartheta\cos^2\varphi$ (Ishimaru, 1997, p. 21). The mean squared field ratio of field $Y^2$ (denoted by $K^2$ in Mätzler, 1998) accounts for the difference in electric field inside the scatterers and the background. This can be represented analytically for small spherical or ellipsoidal scatterers with random orientations as follows (Sihvola, 1999, eq. 4.20):

$$Y^2 = \frac{1}{3}\sum_{j=1}^{3}\left|\frac{\epsilon_a}{\epsilon_a + (\epsilon_2 - \epsilon_1)A_j}\right|^2 \tag{6}$$

where $A_j$ are the depolarization factors along the Cartesian directions. In SMRT version 1.0, only isotropic microstructures are considered which implies $A_j = 1/3$. The apparent permittivity is $\epsilon_a = \frac{1}{3}(2\epsilon_{\text{eff}} + \epsilon_1)$ (Mätzler, 1998). The microstructure term $M(|\mathbf{k}_d|)$ is a function of the difference of wavevectors in the effective medium in the incident and scattering directions, so the modulus is given by:

$$|\mathbf{k}_d| = 2k_0\sqrt{\epsilon_{\text{eff}}}\sin\left(\frac{\Theta}{2}\right) \tag{7}$$

where $\Theta$ is the scattering angle, i.e. the angle between the incident and scattering direction and $\epsilon_{\text{eff}}$ denotes the effective permittivity. This microstructure term can be determined from the Fourier Transform $\widetilde{C}$ of the autocorrelation function of the medium indicator function as (Löwe and Picard, 2015):

$$M(|\mathbf{k_d}|) = \frac{1}{4\pi}\frac{\widetilde{C}(|\mathbf{k}_d|)}{f_2(1-f_2)}. \tag{8}$$

Due to the assumption of isotropy, the Fourier transform of the correlation function $\widetilde{C}(\mathbf{k}_d) = \widetilde{C}(|\mathbf{k}_d|)$ depends only on the magnitude $|\mathbf{k}_d|$ of the scattering vector. Several analytical functions for $\widetilde{C}$ are implemented in SMRT, thus offering different representations of the microstructure to choose from. This is detailed in Section 2.3.

Equations (5) to (8) fully determine the phase function in the 1-2 frame. The 4×4 phase matrix in the principal frame is obtained following the method of Tsang et al. (2007); Ding et al. (2010). Co-polarization phase function matrix elements





can be determined for each $\vartheta$ through calculation of $p_{11} = p_{\text{1-2 frame}}(\vartheta, \varphi = \pi/2)$, and $p_{22} = p_{\text{1-2 frame}}(\vartheta, \varphi = \pi)$ and cross-polarization terms in the 1-2 frame vanish, viz. $p_{12} = p_{21} = 0$. Since the structure of the IBA phase matrix is identical to the phase matrix from Rayleigh and Strong Fluctuation Theory (Tsang et al., 2007), the last two diagonal elements can be estimated as $p_{33} = p_{44} = \sqrt{p_{11}p_{22}}$. Finally, the 4×4 phase matrix $\mathbf{P}$ in the principal frame of the radiative transfer equation

(with $z$ axis normal to the Earth surface) is obtained by rotation (Tsang et al., 2007; Mätzler et al., 2006):

$$\mathbf{P}(\mu, \phi, \mu', \phi') = \begin{bmatrix} P_{11} & P_{12} & P_{13} & 0 \\ P_{21} & P_{22} & P_{23} & 0 \\ P_{31} & P_{32} & P_{33} & 0 \\ 0 & 0 & 0 & P_{44} \end{bmatrix} \tag{9}$$

$$= \begin{bmatrix} \cos^2\alpha & \sin^2\alpha & -\frac{1}{2}\sin 2\alpha & 0 \\ \sin^2\alpha & \cos^2\alpha & \frac{1}{2}\sin 2\alpha & 0 \\ \sin 2\alpha & -\sin 2\alpha & \cos 2\alpha & 0 \\ 0 & 0 & 0 & 1 \end{bmatrix} \cdot \begin{bmatrix} p_{11} & 0 & 0 & 0 \\ 0 & p_{22} & 0 & 0 \\ 0 & 0 & p_{33} & 0 \\ 0 & 0 & 0 & p_{44} \end{bmatrix} \cdot \begin{bmatrix} \cos^2\alpha' & \sin^2\alpha' & \frac{1}{2}\sin 2\alpha' & 0 \\ \sin^2\alpha' & \cos^2\alpha' & -\frac{1}{2}\sin 2\alpha' & 0 \\ -\sin 2\alpha' & \sin 2\alpha' & \cos 2\alpha & 0 \\ 0 & 0 & 0 & 1 \end{bmatrix} \tag{10}$$

where $\alpha$ (respectively $\alpha'$) is the angle of rotation from the 1-2 frame to the incident (respectively scattering) frame. It is related to the incident and scattering zenith and azimuth angles in the principal frame by (Mätzler et al., 2006; Tsang et al., 2007, eq.

3.23):

$$\cos\alpha = \frac{\cos\theta'\sin\theta - \cos\theta\sin\theta'\cos(\phi - \phi')}{\sin\Theta}. \tag{11}$$

The scattering angle $\Theta$ is given by (Mätzler et al., 2006, eq. 3.14) $\cos\Theta = \cos\theta\cos\theta' + \sin\theta\sin\theta'\cos(\phi - \phi')$ so that it follows:

$$\cos^2\alpha = \frac{\left(\cos\theta'\sin\theta - \cos\theta\sin\theta'\cos(\phi - \phi')\right)^2}{1 - \left(\cos\theta\cos\theta' + \sin\theta\sin\theta'\cos(\phi - \phi')\right)^2} \tag{12}$$

The angle $\alpha'$ is obtained by exchanging primed and non-primed angles.

Because the IBA phase matrix in the 1-2 frame is diagonal and the fourth component of the rotation matrix is orthogonal to the three others, the fourth component of the phase matrix in the main frame is also orthogonal to the three others. Except if the full Müller matrix is required by the user, the radiative transfer equation can be solved considering only the three first components, thus reducing the computational cost. This is the way it is implemented in SMRT.

The scattering coefficient $\kappa_s$ is, by definition, calculated from the integration of phase matrix over all incident directions:

$$\kappa_s(\theta, \phi) = \frac{1}{4\pi} \int\limits_{4\pi} d\Omega' \mathbf{P}(\theta, \phi; \theta', \phi'). \tag{13}$$

Taking into account the isotropy of the medium (Tsang et al., 2007), the integral can be computed in the 1-2 frame and yields a diagonal matrix with all elements equal to:

$$\kappa_s = \pi \int\limits_0^\pi [p_{11}(\vartheta) + p_{22}(\vartheta)] \sin\vartheta \, d\vartheta \tag{14}$$





The absorption coefficient $\boldsymbol{\kappa}_{\mathrm{a}}$ is also a diagonal matrix will all the elements equal to:

$$\kappa_a = 2k_0 \Im \sqrt{\epsilon_{\mathrm{eff}}}, \tag{15}$$

where $\Im$ denotes the imaginary part. It is worth noting that Mätzler (1998) and Mätzler and Wiesmann (1999) use a different formulation $\kappa_a = k_0 f_2 \Im \epsilon_2 Y^2$ while the MEMLS code uses equation (15). Both equations are numerically close to each other.

The effective permittivity is not only needed to compute the absorption coefficient but also implicitly to compute the boundary reflection equations (3) and (4) to account for the refraction (Snell law's) and the Fresnel coefficients at the interfaces. The default formulation in SMRT IBA is the Polder and van Santen mixing formula as in Mätzler (1998); Mätzler and Wiesmann (1999). Compared to the classical Maxwell-Garnett formula, it is symmetrical between the scatterers and the background and has been shown to be slightly better for snow (Mätzler, 1996; Sihvola, 1999).

## 2.3   Microstructure representations

Different electromagnetic thoeries use different microstructure representations. In the simplest setting of Rayleigh or independent Mie scattering for a collection of spheres, the microstructure is solely characterized by the sphere radius. The positions of the scatterers are random and uncorrelated meaning that inter-penetration is possible. In DMRT the microstructure is provided in terms of the Fourier transform of the pair-correlation function (Tsang et al., 2000a) and analytical developments have

been mainly given for the Sticky Hard Sphere (SHS) model which is determined by two parameters, the sphere radius and the stickiness $\tau$. In IBA, the microstructure is provided by the ACF as shown in Section 2.2. Analytical expressions of ACF for independent spheres and thin shells are given in Mätzler (1998) and MEMLS proposes a generic exponential function (Mätzler and Wiesmann, 1999) parametrized by the correlation length.

SMRT provides a unified and versatile vision of the microstructure representation. Any microstructure model is defined by

specifying the set of required and optional parameters and by providing, at least for use with IBA, an analytical expression of ACF, either for the real-space form or its Fourier transform (or for both). Though, IBA requires only the Fourier transform, cf. Equation (8), some microstructure models suggested in literature such as the level-cut Gaussian random field model (Ding et al., 2010) are rather based on real space expressions. SMRT handles these cases by automatic Fourier transformation. Due to isotropy, required 3D Fourier transforms can be expressed as 1D Bessel transforms:

$$\widetilde{C}(|\mathbf{k}_d|) = 4\pi \int\limits_0^\infty dr\, r^2 \frac{\sin(k_d r)}{k_d r} C(r) \tag{16}$$

in terms of $k_d = |\mathbf{k}_d|$ which are numerically handled as fast (discrete) sine transforms according to (Lado, 1971).

Overall, the microstructure representation in SMRT closely follows a library concept as commonly employed for small angle scattering software such as Breßler et al. (2015). In version 1.0, five different microstructure models are implemented as a starting point. Some microstructure models are defined by the Fourier transform of the ACF, and some by the real space ACF.





The most convenient characterization of a microstructure is in terms of the Fourier transform of the ACF. Presently the following models are implemented:

$$\text{Exponential:} \quad \widetilde{C}_{\mathrm{ex}}(k_d) = \frac{8\pi l_{\mathrm{ex}}^3 f_2(1-f_2)}{[1+(k_d l_{\mathrm{ex}})^2]^2} \tag{17}$$

$$\text{Teubner-Strey:} \quad \widetilde{C}_{\mathrm{TS}}(k_d) = \frac{8\pi \xi_{\mathrm{TS}}^3 f_2(1-f_2)}{[1+(2\pi\xi_{\mathrm{TS}}/d_{\mathrm{TS}})^2]^2 + 2[1-(2\pi\xi_{\mathrm{TS}}/d_{\mathrm{TS}})^2](k_d\xi_{\mathrm{TS}})^2 + (k_d\xi_{\mathrm{TS}})^4} \tag{18}$$

$\quad$ Independent spheres: $\quad \widetilde{C}_{\mathrm{sph}}(k_d) = f_2(1-f_2)v(a)P(k_d a) \tag{19}$

$\quad\;$ Sticky hard spheres: $\quad \widetilde{C}_{\mathrm{shs}}(k_d) = f_2\, v(a)\, P(k_d a)S_{\mathrm{shs}}(k_d a) \tag{20}$

in terms of the sphere volume $v(a) = 4/3\,\pi\,a^3$, the spherical form factor $P(X)$ defined by

$$P(X) = \left[\frac{3\big(\sin(X) - X\cos(X)\big)}{(X)^3}\right]^2. \tag{21}$$

The SHS structure factor $S_{\mathrm{shs}}$ defined by

$$
\begin{aligned}
\quad S_{\mathrm{shs}}(X) &= \left[A_0(X)^2 + B_0(X)^2\right]^{-1} \\
A_0(X) &= \frac{f_2}{1-f_2}\left[\left(1 - tf_2 + \frac{3f_2}{1-f_2}\right)\Phi(X) + \big(3 - t(1-f_2)\big)\Psi(X)\right] + \cos(X) \\
B_0(X) &= \frac{f_2}{1-f_2}\,X\Phi(X) + \sin(X) \\
\Phi(X) &= 3\left[\frac{\sin(X)}{X^3} - \frac{\cos(X)}{X^2}\right] \\
\Psi(X) &= \frac{\sin(X)}{X}
\end{aligned}
\tag{22}
$$

$\quad$ with $X = k_d a$ and $t$ is given by the smallest solution of the quadratic equation:

$$\frac{f_2}{12}t^2 - \left(\tau + \frac{f_2}{1-f_2}\right)t + \frac{1+f_2/2}{(1-f_2)^2} = 0\,. \tag{23}$$

under the additional condition $t < (1+2f_2)/(f_2(1-f_2))$ which guarantees $S_{\mathrm{shs}}(0)$ to be positive (Baxter, 1968; Tsang and Kong, 2001).

Note, that each microstructure model comes with its own parameters, such as the exponential correlation length $l_{\mathrm{ex}}$ for the 20 $\;$ exponential model, the repeat distance $d_{\mathrm{TS}}$ and the correlation length $\xi_{\mathrm{TS}}$ for the TS model, the sphere radius $a$ for the SPH model, and sphere radius $a$ and stickiness $\tau$ for the SHS model.

The necessity of including also models that are defined via the real space ACF mainly originates from the use of level-cut Gaussian random field models in the context commonly termed bicontinous DMRT (Ding et al., 2010; Chang et al., 2014). To this end we implemented a microstructure model that is defined by

$$\text{Gaussian Random Field:} \quad C_{\mathrm{GRF}}(r) = \frac{1}{2\pi}\int_0^{C_\psi(r)} dt\,\frac{1}{\sqrt{1-t^2}}\exp\left[-\frac{\beta^2}{1+t}\right] \tag{24}$$

$$\text{with } C_\psi(r) = \exp(-r/\xi_{\mathrm{GRF}})\left(1 + \frac{r}{\xi_{\mathrm{GRF}}}\right)\frac{\sin(2\pi r/d_{\mathrm{GRF}})}{(2\pi r/d_{\mathrm{GRF}})}. \tag{25}$$

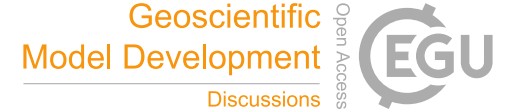



Here $r$ denotes the lag distance from one point to another in the medium. In the case of level-cut Gaussian random fields, the ACF of the bicontinuous medium is determined (Teubner, 1991) by the covariance $C_\psi(r)$ of an underlying zero-mean, unit-variance Gaussian random field $\psi$ from which a two-phase microstructure is obtained by "segmentation" of the continuous field values with threshold $\beta$ (cut-level parameter) which is in one-to-one correspondence with the volume fraction $f_2$. Our particular

choice of the field correlation function $C_\psi$ in equation (25) was motivated by the apparent similarity to the Teubner–Strey model (17). This particular form has e.g. been investigated by Roberts and Torquato (1999) and involves the microstructure parameters $d_{\mathrm{GRF}}$ and $\xi_{\mathrm{GRF}}$, similar to the TS model. However, other choices for the ACF, as e.g. used in Ding et al. (2010) based on a Gamma spectral density, are possible and can be implemented in the future.

For running SMRT with DMRT theory, the SHS microstructure must be selected. In contrast, when using IBA, any of the

above microstructure models can be selected.

## 2.4 Model implementation

The model implementation is highly modular to allow switching between several formulations at each stage of the computation and adding new formulations defined by users. Another feature is the extensive use of default behaviors to facilitate an easy use by beginners but still allow experts to set advanced formulations e.g. for specific investigations or sensitivity studies. The

code is carefully encapsulated, each "science" component (indicated by the orange color Fig. 2 and defined in Section 2.1) is designed as an independent module. Table 1 summarizes the available formulations for each component in version 1.0. Additional modules contain input/output components (green boxes in Fig. 2) and *core* infrastructure components (blue boxes in Fig. 2). Green and blue components do not contain any science and the *core* component should not be modified by the users or scientific developers.

To illustrate the mode of operation of the model it is instructive to relate the instructions of a tiny but fully-functioning code snippet to the model operations carried out in the background:

```python
# import smrt functions
from smrt import make_snowpack, make_model, sensor_list
# create the snowpack
snowpack = make_snowpack(thickness=[100.0],
                         microstructure_model="exponential",
                         density=320.0,
                         temperature=270,
                         corr_length=50e-6)
# create the sensor with the AMSR-E predefined sensor
radiometer = sensor_list.amsre('37V')
# create the model
m = make_model("iba", "dort")
```





```
# run the model
result = m.run(radiometer, snowpack)
# outputs
print(result.TbV(), result.TbH())
```

The user first builds a snowpack by providing the defining properties of each layer, interface and the substrate. Layer characteristics always include density and a microstructure model to use (e.g. microstructure using exponential autocorrelation or sticky hard sphere). The specification of temperature is optional, mostly relevant for the passive mode. Additional parameters depend on the selected microstructure model. For instance, the exponential function requires the exponential correlation length while SHS requires the sphere radius and stickiness. For the interfaces between snow layers, the choice is presently limited to

a "flat interface" which does not require any parameter. In the future rough interfaces could be implemented. The substrate can be selected from various models of soil, a homogeneous medium with flat surface (e.g. bulk of isothermal ice) or a reflector with reflectivity coefficients prescribed by the user.

In the second step, the definition of the model is completed by selecting the electromagnetic theory (that computes the scattering and absorption coefficients, phase matrix, and effective permittivity) and the radiative transfer solver. As mentioned

before, some electromagnetic theories are only compatible with particular microstructure models, e.g. DMRT only works with SHS, Rayleigh works with any microstructure that defines a radius but inherently considers independent spheres. For solving the radiative transfer equation, only the DORT method is currently implemented, based on (Picard et al., 2004, 2013), though two or six-flux solvers (Wiesmann and Mätzler, 1999) could be implemented in the future as well. In the next step the sensor characteristics are specified (active or passive, frequencies, polarizations, ...). For convenience, a list of predefined sensors

is available but sensors with arbitrary characteristics can be defined. The last step is to launch the simulation by combining the prescribed snowpack, the sensor and the defined "model" to obtain a result (e.g. brightness temperature, backscattering coefficient or muller matrix).

The model is implemented in Python (2.7+ and 3.4+) which makes it easy to implement switchable formulations with default and extensible behaviors. This also avoids the cumbersome step of code compilation, though at the cost of a computational

overhead compared to compiled languages. To limit this drawback, the model uses common numerical libraries extensively, such as Numpy and Scipy, allowing fast and numerically accurate calculations. The code is fully documented. It also entirely uses SI units without prefix to avoid any ambiguity.

In addition, we provide different tools for convenience: 1) To facilitate convenient computation of time-series or sensitivity study by a few, clear-cut lines of code the model can be run on lists of different snowpacks 2). To foster comparisons between

SMRT and other common existing models (MEMLS, DMRT-QMS and HUT), we provide language bindings to seamlessly run these models within SMRT, that is using the prescribed snowpack in SMRT and collecting results as if they were produced by SMRT. This requires that the source code of these models is separately installed (they are not distributed with SMRT for licensing reasons). Note that this feature is currently limited to the passive mode.





## 3    Model validation and exploration of the microstructure

As SMRT seeks to unify formulations from other models, a natural starting point for the validation of SMRT is a model comparison (namely with DMRT-ML, DMRT-QMS and MEMLS) to assess the validity of the implementation. This is conducted in Section 3.1. Note that the performance of various models against observations has been extensively evaluated in the past and is not repeated here. Instead, we exploit the fact that SMRT offers the opportunity to compare different microstructure formulations within the same electromagnetic framework to investigate equivalent aspects of different microstructure models. This is addressed in Section 3.2. For the sake of readability, all comparisons are carried out, unless otherwise stated, for a single snowpack-sensor configuration of 37 GHz in a semi-infinite medium. Different configurations can be easily explored by adapting the code that is used to build the figures and provided as open source (see *data availability*).

### 3.1    Comparison with other reference models

#### 3.1.1    The sparse medium approximation

For a sparse medium, i.e. when density tends to zero, many formulations must show the same behavior as the independent spheres with Rayleigh or Mie theory. In SMRT, it is possible to run several combinations of microstructure and electromagnetic models as shown in Figure 3. The results show that at the origin (for $f_2 \rightarrow 0$) the linear trend is the same for several microstructures (independent spheres, non-sticky hard spheres and sticky hard spheres) and different theories (Rayleigh, DMRT QCA-CP, IBA). These results provide a first technical validation of the SMRT implementation of several theories. However, the sparse medium approximation is valid only for very low densities in the range 10-20 kg m$^{-3}$ which is unrealistic for the goal of snow modeling. It is well known that scattering in snow must be treated with dense media theories such as DMRT or IBA. The results from Figure 3 already indicate that the influence of microstructure on deviations from the sparse medium assumption for the scattering coefficient at low densities is more severe than the electromagnetic theory. The next sections therefore consider dense media and a detailed comparison between different microstructure models.

#### 3.1.2    Comparison of SMRT to DMRT-based models

We compare SMRT to results produced from original code of several DMRT variants. Figure 4 shows the angle dependence of the brightness temperature and backscattering coefficient for SMRT DMRT compared to other models for a semi-infinite medium with sphere radius of 0.1 mm, density of 300 kg m$^{-3}$, stickiness of $\tau = 0.5$ and temperature of 256 K. The results reveal that the closest implementation to SMRT DMRT is the model DMRT-ML (Picard et al., 2013). Both use exactly the same formulation for the scattering and absorption coefficient, namely DMRT QCA-CP with small, mono-disperse spheres in the short range approximation (requiring moderate stickiness, i.e. stickiness parameter should not be small). They also use a similar method to solve the radiative transfer equation which explains the small root mean square difference in brightness temperature of about 0.03 K obtained at both polarizations for the angle range 0–60°. In contrast, the comparison of SMRT to DMRT-QMS shows larger differences since the latter computes scattering by DMRT Mie QCA and implements a different





connection of streams between layers in the interface conditions for solving the radiative transfer equation (Picard et al., 2013; Liang et al., 2008). Nevertheless, the differences at both polarizations do not exceed 0.3 K RMS, which is acceptable considering the implementations are different and fully independent. We attribute this difference solely to the RT solvers because we confirmed that running an SMRT simulation with prescribed scattering and absorption coefficients and effective

permittivity pre-computed from DMRT-QMS, the brightness temperature difference of 0.3 K RMS remains unchanged. In active mode (bottom panel in Figure 4), the difference is small as well, 0.65 dB RMS at HH and VV polarizations and 1.4 dB RMS at HV polarization.

The previous results were obtained for small scatterers and moderate stickiness which is compatible with the short range approximation. It is therefore of interest to investigate the limits of this implementation. To this end Figure 5 shows two plots

for the scattering coefficient as a function of sphere radius and stickiness respectively. In the first plot, stickiness is fixed to 0.5 and in the second, the radius is set to 100 μm. DMRT-QMS is considered here as the reference because it implements DMRT QCA Mie which has no theoretical limitations on the size of the particles and on the stickiness parameter. The results show that for radii larger than 185 μm (respectively 285 μm) the error starts to exceed 1 K (respectively 5 K). Translated to SSA values, this corresponds to lower bounds of $17\,\mathrm{m^2kg^{-1}}$ and $11\mathrm{m^2kg^{-1}}$ respectively, which is relatively restrictive for most snow

types (Domine et al., 2007; Roy et al., 2013; Picard et al., 2014), but may be still sufficient for some applications particularly at lower frequencies. Similarly, stickiness values lower than 1 (respectively 0.3) yields an error larger than 1 K (respectively 5 K). Even though stickiness values for natural snow are strictly unknown due to the lack of direct measurements, indirect estimates suggest that values below 1 are common (Löwe and Picard, 2015; Roy et al., 2013). To overcome the restrictive range of validity of DMRT QCA short range, and considering that SMRT version 1.0 does not provide DMRT QCA in the long

range approximation, an alternative strategy is to combine IBA with the SHS microstructure model. Figure 5 shows the results which are much closer to DMRT QMS than DMRT QCA. The difference always remains lower than 5 K in the explored range of input parameters, and it becomes larger than 1K only for radii larger than 285 μm and stickinesses lower than 0.3. This numerical result confirms the quasi-equivalence of the DMRT and IBA theories when using the same microstructure as shown theoretically by (Löwe and Picard, 2015). It even extends this work as only the short range approximation was considered by

Löwe and Picard (2015).

### 3.1.3   Comparison of SMRT to MEMLS-IBA

Figure 6 shows the brightness temperature predicted by SMRT with IBA and MEMLS. For a fair comparison between SMRT and MEMLS, it is required to select the IBA formulation in MEMLS among the 12 available scattering formulations. In addition, MEMLS with IBA allows a choice between different grain shapes which controls the mean squared field ratio $Y^2$.

As SMRT only considers spherical scatterers, MEMLS grain type must be set accordingly (grain type 2 in MEMLS code). The microstructure in SMRT is set to the exponential autocorrelation function as in MEMLS (Mätzler and Wiesmann, 1999) and depends on the correlation length which is set to 100 μm in this computation. The results show a difference of 1.2 K and 1.8 K at V and H polarization respectively. The cause is not the scattering and absorption coefficients which are very close in both models ($\kappa_s = 0.2054\,\mathrm{m^{-1}}$ and $\kappa_a = 0.3092\,\mathrm{m^{-1}}$ for MEMLS and $\kappa_s = 0.2056\,\mathrm{m^{-1}}$ and $\kappa_a = 0.3087\,\mathrm{m^{-1}}$ for SMRT).





Likewise, the effective permittivities are numerically close, 1.5244 in MEMLS and 1.5236 in SMRT which is expected as both uses the same Polder van Santen mixing formula. The difference is thus likely due to the different methods used to solve the radiative transfer. MEMLS uses 6-flux while SMRT uses the DORT method with 32 streams in the simulations presented in this paper. Similar discrepancies were observed when comparing MEMLS to DMRT-ML and DMRT-QMS (fig. 2 Royer et al.,

2017). An implementation of the 6-flux solver in SMRT would provide a route to further explore this issue. It is worth noting that setting a low number of streams in DORT (e.g. 2 or 6) is not recommended and is not equivalent to the 2-flux and 6-flux methods which use specific stream angles and integrals of the bi-static scattering coefficient.

## 3.2  On the equivalence of microstructure models

Equipped with the confidence from the previous sections that SMRT is working as desired, we shall address an actual, open

scientific question. Setting the correct microstructure parameters in microwave model simulations from in-situ observations or snowpack simulations is notoriously difficult and nearly every study uses a different approach. To this end we demonstrate how the equivalence between different approaches can be investigated with SMRT.

The problem originates from the fact that high-level microstructural characterization in terms of the ACF is commonly not available since complete profiles of $\mu$CT or 2D thin sections for the entire snowpack are rare. Instead, density and surface

specific area (SSA) are commonly measured or predicted by snowpack models and the initialization of microwave microstruture models relies on them. The density is unambiguous, the parameter is manifest for each microstructure model and no problems should be expected. In contrast, using SSA is a bit more involved. Theoretically, the SSA is rigorously related to the slope of the ACF at the origin (Debye et al., 1957) and therefore parameterizes a basic size of the constituent scatterers. For microstructures comprising spheres, the SSA [m$^2$ kg$^{-1}$] can be directly converted to sphere radius using $a = 3/(\rho_{\text{ice}} \text{SSA})$. For an exponential

ACF there exists also the well-known relationship (Debye et al., 1957) $l_{\text{ex}} = 4(1 - f_2)/(\rho_{\text{ice}} \text{SSA})$, henceforth termed Debye relation. However most microstructure representations involve three parameters (all except the exponential autocorrelation function) and the additional parameters must be set as well. Although grain type is often observed in the field, quantitative relationships with the microstructure metrics (stickiness or autocorrelation function) have not yet been established and we do not consider this information here.

These issues have been solved in different ways in literature. For the SHS microstructure, Liang et al. (2008) suggest to set the stickiness parameter to "0.1 because it yields 2.8 for the frequency dependence of the extinction coefficient which corresponds to the experimental values (Hallikainen et al., 1987)". These experimental values are the basis of the extinction formulation in the HUT model (Lemmetyinen et al., 2010). However setting stickiness to 0.1 is insufficient to strictly determine the power dependence as it also depends on the grain size and density, i.e. very small scatterers always show a power 4 dependence

(Rayleigh scattering). Another approach was elaborated in a series of empirical studies (Brucker et al., 2010; Picard et al., 2014; Roy et al., 2013; Dupont et al., 2013; Roy et al., 2016). It consists of using non-sticky spheres (i.e. infinite stickiness parameter) and scaling the radius $a$ computed from SSA by an empirical factor $\phi_{\text{SHS}}$ (called "grain size scaling factor"). This factor is obtained by fitting model results to microwave observations. To prevent over-fitting, thereby a single $\phi_{\text{SHS}}$ was applied



to all SSA measurements and the fit was performed using microwave observations at several frequencies, polarizations and/or angles.

To explore if this latter approach is equivalent to choose an optimal stickiness value, we use SMRT to find the equivalent microstructure representations for non-sticky spheres with grain size scaling and sticky spheres. In the following equivalent

microstructures are interpreted as microstructures with same density but different size parameters that produce the same electromagnetic behavior. This is exemplified by using SMRT IBA and matching brightness temperatures at V polarization and $55°$ close to Brewster angle, to integrate properties of scattering and absorption coefficients and phase function (see also Veysoglu and Kong, 1996). Figure 7 shows the grain size scaling factor of non-sticky hard spheres as a function of the stickiness value to obtain this equivalence. For instance $\phi_{SHS} = 2.1$ (used by Picard et al., 2014) is equivalent to a stickiness value around

0.13. Higher values of $\phi_{SHS}$ values up to 3.5 were used in the other studies (Brucker et al., 2010; Roy et al., 2013; Dupont et al., 2013; Roy et al., 2016), corresponding to lower stickiness values approaching 0.1 as suggested by Liang et al. (2008). This confirms that despite using different approaches, these studies converge towards stickiness values in the range $0.1 - 0.2$, in agreement with Löwe and Picard (2015) who retrieved the stickiness from $\mu$CT of snow samples. However, the relationship between stickiness and grain size factor depends on density, especially for $\phi_{SHS} > 2.5$ (Fig. 7), and thus the approach of scaling

grain size cannot be strictly equivalent to selecting an optimal stickiness value.

Though the approach of using a stickiness close to 0.1 seems more physical compared to an empirical scaling factor, it also has weaknesses. Natural snow is composed of grains with variable size, which more resembles a collection of spheres with a distribution of radii (i.e. poly-dispersed spheres). However, the analytical treatment of the ACF for poly-dispersed sticky hard spheres is tedious (Gazzillo et al., 2006) and choosing the distribution form and its parameters is an open question. In the case

of non-sticky small scatterers, Jin (1994) showed that a poly-disperse microstructure can be equivalent to a mono-disperse sphere assembly with an effective radius. This effective radius was found to be about 1.4 times the radius derived from SSA when a Rayleigh distribution of sizes was taken (Jin, 1994). This factor would be slightly different for another distribution but this gives an order of magnitude of the size distribution effect. Based on this, Roy et al. (2013) proposed a pragmatic approach mixing the scaling approach and a fixed stickiness value. For this, they suggested to use $\phi_{SHS} = 1.4$ found by Jin (1994) and

to optimize the stickiness to obtain good fit with observations. This proposition has not been evaluated in other studies.

The exponential autocorrelation is a different and attractive solution because it involves only two parameters that should be fully determined by density and SSA. However, in practice a "hidden" third parameter must be introduced to empirically scale the correlation length in the Debye relation Mätzler (2002). Based on comparisons between simulations and observations, Mätzler (2002) suggested a scaling factor of 0.75 in the Debye relation and justified this adaptation by the necessity of fitting the

exponential function to the real nature of snow, ie. to the actual ACF of snow. However, more recently Montpetit et al. (2013) performed an optimization of the simulations with MEMLS on a large set of observations and found a different coefficient of $1.3 \times 0.75 = 0.975$, suggesting that the Debye relationship was correct without scaling. While the origin of this discrepancy can be understood from the effect of shape (or equivalently size dispersity) of the 3D microstructure (Krol and Löwe, 2016) it remains a practical problem, similar to the freedom of chosing an appropriate stickiness value. To this end we explore

the connection between the Debye scaling factor and stickiness, or in other words, the equivalence between the exponential





ACF with scaled correlation length and SHS. Figure 8 shows the scaling factor $\phi_{\mathrm{exp}}$ of the correlation length in the Debye relationship required to obtain the same electromagnetic behavior as with sticky hard spheres. Each curve is obtained by fixing density and SSA and optimizing $\phi_{\mathrm{exp}}$ to obtain equivalence between exponential and SHS microstructure. The results show that stickiness higher than 0.2 corresponds to $\phi_{\mathrm{exp}}$ lower than 0.5, with little dependence on density. This range seems however

inadequate for snow considering the values of stickiness and $\phi_{\mathrm{exp}}$ used in the literature. Conversely, the value of $\phi_{\mathrm{exp}} = 0.75$ corresponds to stickiness of 0.13 at $200\,\mathrm{kg\,m^{-3}}$, 0.8 at $200\,\mathrm{kg\,m^{-3}}$ and even lower at higher densities. This means that scaling the correlation length proposed by Mätzler (2002) is equivalent to stickiness values suggested by various studies (Liang et al., 2008; Roy et al., 2013). In contrast, $\phi_{\mathrm{exp}} = 1$ is barely accessible for the scaled correlation length derived from the Debye relation indicating the limitations of the exponential ACF for snow. Moreover, the large dependency on density indicates that a

strict equivalence between SHS and an unscaled exponential model is not possible.

The numerical experiments facilitated by SMRT from this section show how different studies, that were hitherto not amenable to a comparison due to apparently different approaches, are now comparable and can be shown to be nearly equivalent for particular parameter choices. Moreover the results unambiguously show that density and SSA are not sufficient to appropriately characterize snow microstructure for microwave modeling purposes and that the sensitivity to a third parame-

ter is highly significant. Until alternative measurement techniques or progress in modeling the microstructure evolution are available, the initialization of microstructure models relies on $\mu$CT characterization or some empiricism to infer the missing parameter.

## 4 Limitations and perspectives

SMRT version 1.0 bears some limitations that are inherent to the architecture as discussed in Section 2.1, others are related to

the current set of available modules and their approximation as shown in Table 1. Some limitations could be simply overcome by implementing new modules or formulations. This section focuses on the latter category.

The scope of SMRT is currently limited to a snowpack over a surface (called substrate) which is a common approach for some applications such as soils, but may be inappropriate for other other snow-covered environments where volume scattering, layering within the substrate or temperature heterogeneity may be important. For instance snow-covered sea-ice or frozen

lakes need to account for bubbly and salty ice with a non-uniform temperature profile. While the generic plane-parallel layered structure in SMRT and the DORT solver are readily suited for this kind of modeling, the electromagnetic behavior of these materials needs to be additionally implemented which is technically easy due to the modular architecture. Bubbly ice (Dupont et al., 2013) has been modeled with DMRT for fresh ice. This should also work for salted ice unless the scattering of brine becomes significant.

Considering soil as a volume scattering medium or accounting for inhomogeneous temperatures or wetness can be treated within DMRT and layered radiative transfer (Lu et al., 2009). Though promising, this approach is still hardly explored yet. Likewise, the atmosphere could benefit from a multi-layer representation as employed in specific, atmospheric radiative transfer models (Eriksson et al., 2011). Implementing atmospheric layers in SMRT would be of interest to deal with cases of strong



surface-atmosphere coupling as observed around 60 GHz near the oxygen absorption band. A simple non-scattering bulk atmosphere can be prescribed in the current SMRT version, however this requires the down and up welling brightness temperatures and transmittance to be calculated externally.

Accurate simulations of snow on the ground in active mode would require more advanced surface scattering models than implemented in the current version. SMRT inherits from the soil modules implemented in DMRT-ML and previously in HUT and MEMLS, which were tailored to the passive mode. These modules mainly compute a specular reflection while a faithful backscatter computation is required for the active mode. DMRT-QMS includes an advanced rough surface treatment from independent numerical simulations (Zhou et al., 2004). In SMRT soil backscatter is prescribed in the current version, but an implementation of a numerical approximate method for rough soil surfaces such as the Advanced Integral Equation Method (AIEM; Chen et al., 2003) is foreseen in the future. Likewise, taking the roughness of the snow surface and internal snow interfaces into account is another interesting perspective (Liang et al., 2009).

A strong assumption in SMRT version 1.0 is the isotropy of the microstructure. Some types of snow have been shown to be highly anisotropic especially due to differences between the vertical and horizontal directions (Löwe et al., 2013). This results in polarization effects in the volume (Leinss et al., 2016). Implementing anisotropic microstructures is possible in the existing architecture but requires significant developments at several locations, namely i) the effective permittivity tensor ii) scattering and absorption coefficients and phase function and iii) solution of the radiative transfer equation taking into account the ordinary and extraordinary streams. Another, related assumption in the current version is the isotropy at the snowpack scale. Accounting for anisotropically reflecting interfaces would only require an improvement of the radiative transfer solver and the implementation anisotropic surface reflections. However, to include all emergent effects (such as multiple scattering between surface and volume) a full 3D model is required, which is not compatible with the SMRT architecture.

Some limitations of SMRT are inherent to the radiative transfer equation which does not keep track of the absolute phase. This obviously prevents interferometric calculations and may be restrictive when the layer thickness is smaller than the wavelength of the microwaves, that is, at low frequencies (e.g. at L, band, Tan et al., 2015a; Leduc-Leballeur et al., 2015) or in case of thin ice lenses in the snowpack (Mätzler, 1987). In some cases, ad hoc corrections of the radiative transfer solution can be implemented. For instance MEMLS (Wiesmann and Mätzler, 1999) computes the effect of constructive interferences between the interfaces of sub-wavelength layers at the condition that scattering is negligible and these thin layers are surrounded by thick layers. This correction is suitable for isolated ice lenses (Montpetit et al., 2013) but not sufficient for low frequencies. Another important case concerns the active mode in the backscatter direction – which is the most common configuration for radars. In such a configuration, some of the many possible trajectories of radiation propagation are paired, cyclical double bounces involving reflections between one of the interfaces (soil or air-snow surface) and the volume. Theses pairs constructively interfere with each other, according to wave theory. As a result, the backscatter for these bounces is increased by 3dB compared to what is predicted by the incoherent radiative transfer theory. This phenomenon called backscattering enhancement has recently been taken into account by developing a specific solver of the radiative transfer equation able to distinguish the non-cyclical and cyclical trajectories, and to apply a correction of 3 dB to the latter group (Tan et al., 2015b).



Another limitation concerns simulations of altimetric signal or FMCW radar. The radiative transfer equation solver available in SMRT version 1.0 considers the stationary radiative transfer equation (1) which is insufficient to simulate altimetry waveform or time-resolved radar echo. However, the SMRT architecture could accommodate such an enhancement with little change, only an adequate solver needs to be added (e.g. Lacroix et al., 2008).

Finally we acknowledge that the Python implementation of SMRT bears some peculiarities. By extensively using Python dynamic capabilities, the model computation is probably less efficient than specialized code, even though numerically critical code is delegated to optimized libraries through SciPy. Because of Python, the model may be inadequate for high performance computation. In this case SMRT may be still useful for prototyping and determining the optimal subset of formulations that could then be implemented in compiled language since a numerical reference greatly helps to achieve such an optimization

step. Moreover, it is worth noting that the Python ecosystem for high performance computing is fast improving and that SMRT code may be parallelized in the future.

## 5    Conclusions

A new radiative transfer model to simulate emission or radar echo from a snowpack has been presented in this paper. It is built around the radiative transfer equation and specifically tailored to model snow but in the future also other plane-parallel

media in the cryosphere. SMRT differs from other models in its scope in many aspects. SMRT is not a new model with a more advanced theory, it is rather a repository of established formulations or widely-used model configurations that can be easily inter-changed. The novelty is thus to allow testing different existing configurations and explore new ones, in particular regarding the microstructure. Using SMRT, we have highlighted the equivalence between different widely-used microstructure representations (SHS and exponential autocorrelation function) and different approaches proposed in the literature to run

simulations based on in-situ measurements. These results show that to fully describe snow in microwave models requires at least three main metrics, the density, grain size, and another parameter characterizing larger scale structural correlations of the ice matrix. The fact that these latter properties are presently inaccessible by other measurements or snowpack modeling contributes to the uncertainties in microwave simulations, and actually constitutes one of biggest challenges to solve.

The numerical validation of SMRT has shown the numerically equivalence with DMRT-ML for the DMRT QCA-CP electro-

magnetic formulation and has shown close results with DMRT-QMS under DMRT QCA under the small scatterer assumption in passive and active mode even though small differences remain unexplained. Larger differences are observed with respect to MEMLS, which we attribute to the six-flux method used by MEMLS to solve the radiative transfer equation. Regarding HUT, SMRT contains no sufficiently similar configuration to perform a validation. Nevertheless the language binding to the HUT code have been included for future comparisons with other configurations. Not all SMRT configurations and available

microstructure representations have been tested in this study because of the large number of possible combinations, this is left to future work.

Several limitations of SMRT version 1.0 have been outlined that can be readily overcome by model extensions which are supported by modularity. The developed code is highly structured for each step of the radiative transfer calculation. The





model is designed to facilitate future developments of existing and new formulations without changing existing code, which should foster community-based contributions and consolidate SMRT as a repository of the community knowledge. Future work includes implementation of new features to account for different media (e.g. sea-ice), variants of electromagnetic models (e.g. DMRT QCA long range) or radiative transfer solver (e.g. six-flux solver or time-resolved radiative transfer equation) to
increase the scope of applications. In this paper we focused on two widely used microstructure representations; SMRT already includes other representation and new ones could be included, such as empirical autocorrelation functions derived from $\mu$CT, which opens a new promising way to characterize the microstructure.

## 6   Code availability

The code is released under the LGPLv3 open source library and is available from https://github.com/smrt-model/smrt.

## 7   Data availability

The jupyter notebooks to generate the figures are available from https://github.com/smrt-model/smrt1paper

## 8   Appendix A: DORT method

The discrete ordinate and eigenvalue method is a widely used method to solve the radiative transfer equation for multi-layered media. It is particularly recommended when optical depth is thick and multiple scattering is significant (Liu et al., 2016).
Overall, it is not the most computationally efficient but is robust. Many variants have been proposed in the literature for the scalar or vector (i.e. polarized) radiative transfer equations, for dense or sparse media, for passive mode only, and with different basis functions for the azimuthal angle (Liu et al., 2016). DORT in SMRT inherits from the passive-only variant used in DMRT-ML (Jin, 1994; Picard et al., 2013) and the active-only sparse medium variant developed in Picard et al. (2004) for the simulation of forest backscatter. It is also similar to the DORT method implemented in DMRT-QMS and only differ in the way
the interface conditions are handled. In SMRT the streams in the different layers are directly connected and as a consequence their zenith angles governed by the Snell refraction law whereas DMRT-QMS uses constant angles and spline interpolation to connect the streams.

The transformation of the radiative transfer equation into a system of linear ordinary differential equations requires the discretization of the azimuthal and zenith angular dependences. The $\phi$ dependence is treated by decomposition into cosine and
sine functions:

$$\mathbf{I}(\mu, \phi, z) = \sum_{m=0}^{\infty} \mathbf{I}^{c,m}(\mu, z) \cos(m\phi) + \mathbf{I}^{s,m}(\mu, z) \sin(m\phi) \tag{26}$$



with Fourier coefficients $\mathbf{I}^{c,m}$ and $\mathbf{I}^{s,m}$. Because of the azimuthal symmetry of the medium, the elements of the intensity vector are either purely even or odd functions of $\phi$ so that:

$$\mathbf{I}^{c,m} = \begin{bmatrix} I_v^{c,m} \\ I_h^{c,m} \\ 0 \\ 0 \end{bmatrix} \tag{27}$$

and

$$\mathbf{I}^{s,m} = \begin{bmatrix} 0 \\ 0 \\ U^{s,m} \\ V^{s,m} \end{bmatrix} \tag{28}$$

Similarly the phase matrix writes:

$$\mathbf{P}^{(l)}(\mu,\phi,\mu',\phi') = \sum_{m=0}^{\infty} \mathbf{P}^{c,(l),m}(\mu,\mu')\cos[m(\phi-\phi')] + \mathbf{P}^{s,(l),m}(\mu,\mu')\sin[m(\phi-\phi')] \tag{29}$$

and because of the azimuthal symmetry of the medium, some elements of the phase matrix components are even functions of $\phi$ so that:

$$\mathbf{P}^{c,m} = \begin{bmatrix} P_{11}^{c,m} & P_{12}^{c,m} & 0 & 0 \\ P_{21}^{c,m} & P_{22}^{c,m} & 0 & 0 \\ 0 & 0 & P_{33}^{c,m} & P_{34}^{c,m} \\ 0 & 0 & P_{43}^{c,m} & P_{44}^{c,m} \end{bmatrix} \tag{30}$$

and other ones are odd functions:

$$\mathbf{P}^{s,m} = \begin{bmatrix} 0 & 0 & P_{13}^{s,m} & P_{14}^{s,m} \\ 0 & 0 & P_{23}^{s,m} & P_{24}^{s,m} \\ P_{31}^{s,m} & P_{32}^{s,m} & 0 & 0 \\ P_{41}^{s,m} & P_{42}^{s,m} & 0 & 0 \end{bmatrix} \tag{31}$$

By inserting (26) and (29) into equation (2), multiplying by $\cos(m\phi)$ and integrating over $\phi$ from 0 to $2\pi$, we obtain:

$$\begin{aligned}
\mu \frac{d\mathbf{I}^{c,m}(\mu,z)}{dz} = & -\boldsymbol{\kappa}_e^{(l)}(\mu)\mathbf{I}^{c,m}(\mu,z) \\
& + \int_{-1}^{1} d\mu' \left[ \mathbf{P}^{c,(l),m}(\mu,\mu')\mathbf{I}^{c,m}(\mu',z) - \mathbf{P}^{s,(l),m}(\mu,\mu')\mathbf{I}^{s,m}(\mu',z) \right] \\
& + \delta_m \boldsymbol{\kappa}_a^{(l)}(\mu)T^{(l)}\mathbf{1}
\end{aligned} \tag{32}$$

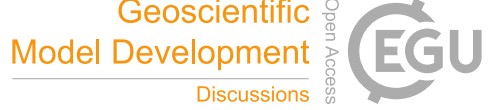

and

$$\mu \frac{d\mathbf{I}^{\mathrm{s},m}(\mu,z)}{dz} = -\boldsymbol{\kappa}_e^{(l)}(\mu)\mathbf{I}^{\mathrm{s},m}(\mu,z)$$

$$+ \int_{-1}^{1} d\mu' \left[ \mathbf{P}^{\mathrm{s},(l),m}(\mu,\mu')\mathbf{I}^{\mathrm{c},m}(\mu',z) + \mathbf{P}^{\mathrm{c},(l),m}(\mu,\mu')\mathbf{I}^{\mathrm{s},m}(\mu',z) \right]$$

$$+ \delta_m \boldsymbol{\kappa}_a^{(l)}(\mu)T^{(l)}\mathbf{1} \tag{33}$$

where $\delta_m$ is 1 for $m = 0$ and 0 otherwise. We then introduce the even intensity and phase matrix:

$$\mathbf{I}^{\mathrm{e},m} = \begin{bmatrix} I_{\mathrm{V}}^{\mathrm{c},m} \\ I_{\mathrm{H}}^{\mathrm{c},m} \\ U^{\mathrm{s},m} \\ V^{\mathrm{s},m} \end{bmatrix} \tag{34}$$

and

$$\mathbf{P}^{\mathrm{e},(l),m} = \begin{bmatrix} P_{11}^{\mathrm{c},(l),m} & P_{12}^{\mathrm{c},(l),m} & -P_{13}^{\mathrm{s},(l),m} & -P_{14}^{\mathrm{s},(l),m} \\ P_{21}^{\mathrm{c},(l),m} & P_{22}^{\mathrm{c},(l),m} & -P_{23}^{\mathrm{s},(l),m} & -P_{24}^{\mathrm{s},(l),m} \\ P_{31}^{\mathrm{s},(l),m} & P_{32}^{\mathrm{s},(l),m} & P_{33}^{\mathrm{c},(l),m} & P_{34}^{\mathrm{c},(l),m} \\ P_{41}^{\mathrm{s},(l),m} & P_{42}^{\mathrm{s},(l),m} & P_{43}^{\mathrm{c},(l),m} & P_{44}^{\mathrm{c},(l),m} \end{bmatrix} \tag{35}$$

The respective odd intensity and phase matrix contributions vanish because of the azimuthal symmetry. The RT equation for the even vector in each layer $l$ reads:

$$\mu \frac{d\mathbf{I}^{\mathrm{e},m}(\mu,z)}{dz} = -\boldsymbol{\kappa}_e^{(l)}(\mu)\mathbf{I}^{\mathrm{e},m}(\mu,z) \quad + \int_{-1}^{1} d\mu' \left[ \mathbf{P}^{\mathrm{e},(l),m}(\mu,\mu')\mathbf{I}^{\mathrm{e},m}(\mu',z) \right] + \delta_m \boldsymbol{\kappa}_a^{(l)}(\mu)T^{(l)}\mathbf{1}. \tag{36}$$

In the following the superscript e is dropped.

The $\theta$ dependence is replaced by a set of discrete ordinates (samples at fixed angles), so that the integral over $\mu'$ is transformed into a discrete sum as:

$$\int_{-1}^{1} d\mu' \mathbf{P}^{(l),m}(\mu,\mu')\mathbf{I}^m(\mu',z) \approx$$

$$\sum_{i=1}^{N(l)} w_i^{(l)} \left[ \mathbf{P}^{(l),m}(\mu,\mu_i^{(l)})\mathbf{I}^m(\mu_i^{(l)},z) + \mathbf{P}^{(l),m}(\mu,-\mu_i^{(l)})\mathbf{I}^m(-\mu_i^{(l)},z) \right] \tag{37}$$

where $\mu_i^{(l)}$ and $-\mu_i^{(l)}$ are the cosines where the integrand is evaluated in layer $l$ (the points are symmetric around 0 and by
convention $\mu_i^{(l)}$ is positive). $w_i^{(l)}$ are the corresponding weights. It is worth noting that the number of samples $N(l)$ and their positions differ for each layer depending on the refraction index. Introducing the matrix/vector notation for the linear solver, we define an intensity vector $\boldsymbol{\mathcal{I}}^m$ containing the four Stokes components (or a subset of them) for all directions, that is with





cosine $\mu$ ranging from $\mu_1^{(l)}$ to $\mu_{N(l)}^{(l)}$ and then from $-\mu_1^{(l)}$ to $-\mu_{N(l)}^{(l)}$. This vector has $2 \times 4 \times N(l)$ elements. The extinction matrix $\boldsymbol{\kappa}_e^{(l)}$ and the phase matrix $\boldsymbol{\mathcal{P}}^{(l),m}$ are defined similarly. These matrices contain $2 \times 4 \times N(l)$ rows and columns. Finally, we define the weight diagonal matrix $\mathbf{w}^{(1)}$ containing weights $w_i$, $i = 1 \ldots N(l)$ for each Stokes components and upward and downward directions. Similarly we define the cosine diagonal matrix $\mu^{(l)}$ containing the $\mu_i^{(l)}$ elements, $i = 1 \ldots N(l)$ followed

by the $-\mu_i^{(l)}$ elements. Then, by applying (32) in the discrete directions $\mu_i^{(l)}$ and $-\mu_i^{(l)}$, $i = 1, \ldots N(l)$, the discrete RT equation for mode $m$ within layer $l$ is obtained:

$$\frac{d\boldsymbol{\mathcal{I}}^{(l),m}(z)}{dz} = -\boldsymbol{\mathcal{A}}^{(l),m}\boldsymbol{\mathcal{I}}^{(l),m}(z) + \delta_m \mu^{(l)^{-1}} \boldsymbol{\kappa}_a^{(l)} T^{(l)} \mathbf{1} \tag{38}$$

where

$$\boldsymbol{\mathcal{A}}^{(l),m} = [\mu^{(l)^{-1}} \boldsymbol{\kappa}_e^{(l)} - \mu^{(l)^{-1}} \boldsymbol{\mathcal{P}}^{(l),m} \mathbf{w}] \tag{39}$$

This is a non-homogeneous system of ordinary first order differential equations with constant coefficients.

## 8.1   General solution of the discrete RT equation within a layer

Within each layer $l$ ($l = 1, \ldots L$), diagonalisation of the $\boldsymbol{\mathcal{A}}^{(l)}$ matrix yields $8N(l)$ eigenvalues $\beta_j^{(l),m}$ and their corresponding eigenvectors $\boldsymbol{\mathcal{E}}_j^{(l),m}$:

$$\boldsymbol{\mathcal{A}}^{(l),m} \boldsymbol{\mathcal{E}}_j^{(l),m} = \beta_j^{(l),m} \boldsymbol{\mathcal{E}}_j^{(l),m} \qquad j = 1, \ldots 8N(l). \tag{40}$$

The general solution of the discrete equation (38) in layer $l$ can then be written:

$$\boldsymbol{\mathcal{I}}^{(l),m}(z) = \boldsymbol{\mathcal{E}}^{(l),m} \boldsymbol{\mathcal{D}}^{(l),m}\left(z - z_0^{(l)}(\mu)\right) \boldsymbol{x}^{(l),m} + \delta_m T^{(l)} \mathbf{1} \tag{41}$$

where $\boldsymbol{\mathcal{E}}^{(l),m}$ is the matrix containing the eigenvectors $\boldsymbol{\mathcal{E}}_j^{(l),m}$ as columns, $\boldsymbol{\mathcal{D}}^{(l),m}(z) = \mathrm{diag}(e^{-\beta_1^{(l),m}z} \ldots e^{-\beta_{4N(l)}^{(l),m}z})$ is the diagonal matrix describing the transmission of eigenvectors through the layer and $\boldsymbol{x}^{(l),m}$ are the unknown constants to be determined from the boundary conditions. For the purpose of the boundary conditions we distinguish the upwelling ($\mu > 0$) and downwelling ($\mu < 0$) sub-spaces. In addition for the numerical stability, the reference height is chosen at the top of

the layer for the downwelling $z_0^{(l)}(\mu < 0) = z_{l-1}$ and at bottom for the upwelling $z_0^{(l)}(\mu > 0) = z_l$. Hence the elements of $\boldsymbol{\mathcal{D}}^{(l),m}\left(z - z_0^{(l)}(\mu)\right)$ are always between 0 and 1.

## 8.2   The boundary conditions

At this point, the problem consists of determining the $8N(l)$ unknown constants $\boldsymbol{x}^{(l),m}$ per layer, i.e., $4\sum_{l=1}^{L} N(l)$ unknown

constants in total. The boundary condition stems from the necessary energy conservation at the interfaces and depends on the bistatic reflection coefficient at the interface (for both upwelling and downwelling waves). The precise form of BC depends on the choice of cosines $\mu_i^{(l)}$ in the layers. Here, we make a specific choice following the strategy used in DMRT-ML and described in Picard et al. (2013). The cosines in the layer with highest refractive index (most refringent) is taken to follow the

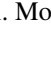



(optimal) Gaussian quadrature rules, and the cosine in the other layers is deduced based on Snell's refraction law. This leads to a one-to-one connection between the streams in adjacent layers, except in the case of total reflection where the more grazing streams in the more refringent layers are not connected to any stream in their less refringent neighbor layers (Picard et al., 2013). This choice yields a simple form of the boundary conditions. The downwelling waves at the top interface of a layer $l$ is

the sum of the reflected upwelling waves of layer $l$ and the downwelling transmitted waves of layer $l-1$:

$$\mathcal{I}_{\mu<0}^{(l),m}(z_{l-1}) = \mathcal{R}^{\text{top},(l),m}\mathcal{I}_{\mu>0}^{(l),m}(z_{l-1}) + \mathcal{T}^{\text{bottom},(l-1)m}\mathcal{I}_{\mu<0}^{(l-1)m}(z_{l-1}) \tag{42}$$

where $\mathcal{R}^{\text{top},(l),m} = \left(\mathcal{R}^{\text{spec,top},(l),m} + \mathcal{R}^{\text{diff,top},(l),m}\mathbf{w}\right)$ and similarly for the other $\mathcal{R}$ and $\mathcal{T}$ matrices. The upwelling waves at the bottom interface of a layer $l$ is the sum of the reflected downwelling waves of layer $l$ and the upwelling transmitted waves of layer $l+1$:

$$\mathcal{I}_{\mu>0}^{(l),m}(z_l) = \mathcal{R}^{\text{bottom},(l),m}\mathcal{I}_{\mu<0}^{(l),m}(z_l) + \mathcal{T}^{\text{top},(l+1)m}\mathcal{I}_{\mu>0}^{(l+1+)m}(z_l). \tag{43}$$

The $\mathcal{R}$ and $\mathcal{T}$ matrices are described and discretized in a similar manner to the phase matrix. Inserting Eq. (41) into Eq. (42),(43) yields:

$$\begin{aligned}
\mathcal{E}_{\mu<0}^{(l),m}\mathcal{D}_{\mu<0}^{(l),m}\left(z_{l-1} - z_0^{(l)}(\mu<0)\right)\boldsymbol{x}_{\mu<0}^{(l),m} = &\ \mathcal{R}^{\text{top},(l),m}\mathcal{E}_{\mu>0}^{(l),m}\mathcal{D}_{\mu>0}^{(l),m}\left(z_{l-1} - z_0^{(l)(\mu>0)}\right)\boldsymbol{x}_{\mu>0}^{(l),m} \\
&+ \mathcal{T}^{\text{bottom},(l-1)m}\mathcal{E}_{\mu<0}^{(l-1)m}\mathcal{D}_{\mu<0}^{(l-1)m}\left(z_{l-1} - z_0^{(l-1)}(\mu<0)\right)\boldsymbol{x}_{\mu<0}^{(l-1)m} \\
&+ \mathcal{T}^{\text{bottom},(l-1)m}\delta_m T^{(l-1)} - \left(1 - \mathcal{R}^{\text{top},(l),m}\right)\delta_m T^{(l)}
\end{aligned} \tag{44}$$

and

$$\begin{aligned}
\mathcal{E}_{\mu>0}^{(l),m}\mathcal{D}_{\mu>0}^{(l),m}(z_l - z_0^{(l)})\boldsymbol{x}_{\mu>0}^{(l),m} = &\ \mathcal{R}^{\text{bottom},(l),m}\mathcal{E}_{\mu<0}^{(l),m}\mathcal{D}_{\mu<0}^{(l),m}\left(z_l - z_0^{(l)}(\mu<0)\right)\boldsymbol{x}_{\mu<0}^{(l),m} \\
&+ \mathcal{T}^{\text{top},(l+1)m}\mathcal{E}_{\mu>0}^{(l+1)m}\mathcal{D}_{\mu>0}^{(l+1)m}\left(z_l - z_0^{(l+1)}(\mu>0)\right)\boldsymbol{x}_{\mu>0}^{(l-1)m} \\
&+ \mathcal{T}^{\text{top},(l+1)m}\delta_m T^{(l+1)} - \left(1 - \mathcal{R}^{\text{bottom},(l),m}\right)\delta_m T^{(l)}
\end{aligned} \tag{45}$$

where the first equation applies to $l > 1$ and the last to $l < L - 1$.

The additional boundary condition for the bottommost layer $l = L$ is:

$$\begin{aligned}
\mathcal{E}_{\mu>0}^{(l),m}\mathcal{D}_{\mu>0}^{(l),m}(z_l - z_0^{(l)})\boldsymbol{x}_{\mu>0}^{(l),m} = &\ \mathcal{R}^{\text{bottom},(l),m}\mathcal{E}_{\mu<0}^{(l),m}\mathcal{T}_{\mu<0}^{(l),m}(z_l - z_0^{(l)})\boldsymbol{x}_{\mu<0}^{(l),m} \\
&+ \mathcal{T}^{\text{top},(l+1)m}\delta_m T^{(l+1)} - \left(1 - \mathcal{R}^{\text{bottom},(l),m}\right)\delta_m T^{(l)}
\end{aligned} \tag{46}$$

considering that $l = L + 1$ designates the substrate.

And finally, using (43), the condition for the air-snow interface ($l = 1$) yields:

$$\begin{aligned}
\mathcal{E}_{\mu<0}^{(l),m}\mathcal{D}_{\mu<0}^{(l),m}(z_{l-1} - z_0^{(l)})\boldsymbol{x}_{\mu<0}^{(l),m} = &\ \mathcal{R}^{\text{top},(l),m}\mathcal{E}_{\mu>0}^{(l),m}\mathcal{D}_{\mu>0}^{(l),m}(z_{l-1} - z_0^{(l)})\boldsymbol{x}_{\mu>0}^{(l),m} \\
&+ \mathcal{T}^{\text{bottom},(0)m}\mathcal{I}_{\mu<0}^{(0)m}(z_{l-1}) \\
&- \left(1 - \mathcal{R}^{\text{top},(l),m}\right)\delta_m T^{(l)}
\end{aligned} \tag{47}$$



considering that $l = 0$ designates the air above the snowpack.

In active mode, the downwelling beam with incidence angle $\theta_0$ is represented by:

$$\left\{\boldsymbol{\mathcal{I}}_{\mu<0}^{(0)m}(z_{l-1})\right\}_i = \left(\frac{\delta_{ii_0}\alpha}{w_{i_0}} + \frac{\delta_{ii_0-1}(1-\alpha)}{w_{i_0-1}}\right)\frac{1}{(1+\delta_m)\pi}\mathbf{I}_0 \tag{48}$$

where $i = 1 \ldots N(l)$ and the bracket shall be interpreted as taking the coordinates of the vector. Since $\theta_0$ is in general not

exactly on a stream angle, we perform a linear interpolation between the two nearest streams as follows: $i_0$ is the (integer) index so that $\cos\theta_0$ is between $\mu_{i_0}$ and $\mu_{i_0-1}$, then $\alpha = (\cos\theta_0 - \mu_{i_0})/(\mu_{i_0} - \mu_{i_0-1})$. In active mode, except if the fourth Stokes component of the incident beam is non-zero, this component remains null under the isotropic assumption used here. As a consequence only the first three components need to be computed in practice. In the passive microwave mode the situation is even simpler. The incident energy comes from the atmosphere (downwelling atmosphere emission), which is considered

azimuthally symmetrical, so that only the $m = 0$ mode is non-zero and as both the forcing (incident radiative) and the source (emission) have an azimuthal symmetry, the $m > 0$ mode equations all have a zero solution. As a consequence for passive microwave, only the $m = 0$ mode needs to be solved and only the first two components of Stokes vector are non-zero. The SMRT DORT code is built to accommodate a variable number of Stokes components which allow optimized computations.

All boundary conditions provide $4\sum_{l=1}^{L} N(l)$ equations linking the unknowns $\boldsymbol{x}^{(l),m}$ for each of the $L$ layers since the

boundary conditions only connect successive layers, the system of equations takes the form of an almost block diagonal matrix. Picard et al. (2014) uses a Almost Block Diagonal algorithm to solve it whereas DMRT-ML and SMRT cast it as a band diagonal system for which efficient algorithms exists in Lapack and SciPy. Solving this system yield $\boldsymbol{x}^{(l),m}$ for each layer $l$ and each mode $m$. The outgoing intensity from the top layer $l = 1$ can be calculated by inserting $\boldsymbol{x}^{(1)m}$ into the general solution given by (41) and using the topmost boundary conditions (43):

$$\boldsymbol{\mathcal{I}}^{(0)m}(z_0)_{\mu>0} = \boldsymbol{\mathcal{R}}^{\text{bottom},(0)m}\boldsymbol{\mathcal{I}}^{(0)m}(z_0)_{\mu<0} + \boldsymbol{\mathcal{T}}^{\text{top},(1)m}\boldsymbol{\mathcal{I}}^{(1)m}(z_0)_{\mu>0} \tag{49}$$

where

$$\boldsymbol{\mathcal{I}}^{(1)m}(z_0)_{\mu>0} = \boldsymbol{\mathcal{E}}^{(1)m}\boldsymbol{\mathcal{T}}^{(1)m}(z - z_0^{(1)})\boldsymbol{x}^{(1)m} + \delta_m T^{(1)} \tag{50}$$

Last operation is to reconstruct the azimuth series, as follows:

$$\boldsymbol{\mathcal{I}}(\phi) = \sum_{m=0}^{M} \boldsymbol{\mathcal{I}}^{(0)m}(z_0)_{\mu>0}\cos(m\phi) \tag{51}$$

For the passive microwave case, only the first mode $m = 0$ is non-null. For the active microwave case, this equation gives the total outgoing intensity from the snowpack which accounts for both diffuse and the reflected coherent radiations. The backscatter only includes diffuse radiation and in principle the coherent radiation is only in the forward direction. However, the truncation of the $\phi$-series at mode $m$ contains spurious remainders of the coherent intensity in any $\phi$ direction. To overcome this numerical problem the total intensity (51) is written as diffuse and coherent components (Ishimaru, 1997):

$$\boldsymbol{\mathcal{I}}(\phi) = \boldsymbol{\mathcal{I}}^{[d]}(\phi) + \boldsymbol{\mathcal{I}}^{[c]}(\phi) \tag{52}$$





and the diffuse part is obtained by subtracting the total intensity from the coherent part. The former is the result of equation (51) while the latter is computed using the equations as for the total intensity except all sources of scattering are switched off (Picard et al., 2004). This includes volume scattering ($\kappa_s = 0$ and $\boldsymbol{\mathcal{P}}^{(l),m} = 0$) and diffuse reflection ($\boldsymbol{\mathcal{R}}^{\mathrm{diff,top/bottom},(l),m} = 0$). Denoting with a superscript $[c]$ the unknowns and eigenvectors obtained by this calculation, the diffuse intensity is then given by:

$$\boldsymbol{\mathcal{I}}^{[d]}(\phi) = \sum_{m=0}^{M} [\boldsymbol{\mathcal{E}}^{(1)m}\boldsymbol{\mathcal{T}}^{(1)m}(z - z_0^{(1)})\boldsymbol{x}^{(l),m} - \boldsymbol{\mathcal{E}}^{(1)m[c]}\boldsymbol{\mathcal{T}}^{(1)m[c]}(z - z_0^{(1)})\boldsymbol{x}^{(1)[c]m}]\cos(m\phi)$$

The intensity in backscatter direction is finally obtained by setting $\phi = \pi$ and by linear interpolation of the intensities $\boldsymbol{\mathcal{I}}^{[d]}(\phi)$ at $\mu_i^{(0)}$ to the exact incident angle.

## 9 Appendix B: Independent Rayleigh scatterers

This well-known approximation is mostly given for reference as it only applies to sparse medium. In this case, the effective permittivity is equal to that of the background:

$$\epsilon_{\mathrm{eff}} = \epsilon_1 \tag{53}$$

and the scattering coefficient is given by (p 128 Tsang et al., 2000b):

$$\kappa_s = 2k_0^4 a^3 f_2 \left| \frac{\epsilon_2 - \epsilon_1}{\epsilon_2 + 2\epsilon_1} \right|^2, \tag{54}$$

the absorption coefficient by:

$$\kappa_a = 9k_0 f_2 \frac{\Im\epsilon_2}{\epsilon_1} \left| \frac{\epsilon_1}{\epsilon_2 + 2\epsilon_1} \right|^2 \tag{55}$$

and the phase matrix by:

$$\mathbf{P}(\mu, \phi, \mu', \phi') = \begin{bmatrix} f_{vv}^2 & f_{vh}f_{hv} & f_{vh}f_{vv} & 0 \\ f_{hv}^2 & f_{hh}^2 & 2f_{hv}f_{hh} & 0 \\ 2f_{vv}f_{hv} & 2f_{vh}f_{hh} & f_{vv}f_{hh} + f_{vh}f_{hv} & 0 \\ 0 & 0 & 0 & f_{vv}f_{hh} - f_{vh}f_{hv} \end{bmatrix} \tag{56}$$

$$\tag{57}$$

where $f_{vv} = \mu\mu'\cos(\phi - \phi') + \sqrt{(1 - \mu^2)(1 - \mu'^2)}$, $f_{hh} = \cos(\phi - \phi')$, $f_{hv} = -\mu'\sin(\phi - \phi')$ and $f_{vh} = \mu'\sin(\phi - \phi')$.

## 10 Appendix C: DMRT QCA and QCA-CP in the short range approximation

Formulations for DMRT QCA and QCA-CP are available in many studies and briefly recalled here for completeness for the mono-disperse and under the short range approximation. The QCA-CP version is formulated according to Shih et al. (1997).



The first order effective dielectric constant $\epsilon_{\text{eff},0}$ is obtained by solving the following quadratic equation (equation 3 in Shih et al. (1997) with $a = 0$ or equation 5.3.125 in Tsang and Kong (2001)):

$$\epsilon_{\text{eff},0}^2 + \epsilon_{\text{eff},0}\left(\frac{\epsilon_2 - \epsilon_1}{3}(1 - 4f_2) - \epsilon_1\right) - \epsilon_1\frac{\epsilon_2 - 1}{3}(1 - f_2) = 0, \tag{58}$$

where $f_2$ is the fractional volume of scatterers, $\epsilon_1$ and $\epsilon_2$ are the dielectric constants of the background and scatterers, respectively. The effective dielectric constant with scattering Shih et al. (equation 3 in 1997) combined with Equation (58) yields:

$$\epsilon_{\text{eff}} = \varepsilon_1 + (\epsilon_{\text{eff},0} - \varepsilon_1)\left(1 + \jmath\frac{2}{9}\sqrt{\epsilon_{\text{eff},0}}(k_0 a)^3 \frac{\varepsilon_2 - \varepsilon_1}{1 + \frac{\varepsilon_2 - \varepsilon_1}{3\epsilon_{\text{eff},0}}(1 - f_2)}\frac{(1 - f_2)^4}{(1 + 2f_2 - tf_2(1 - f_2))^2}\right), \tag{59}$$

where $a$ is the radius of the spheres and $k_0 = 2\pi/\lambda$ is the wavenumber with $\lambda$ the wavelength. The parameter $t$ is zero for non-sticky spheres and otherwise given by the largest solution of (equation 6 in Shih et al., 1997):

$$\frac{f_2}{12}t^2 - (\tau + \frac{f_2}{1 - f_2})t + \frac{1 + f_2/2}{(1 - f_2)^2} = 0 \tag{60}$$

where $\tau$ is the stickiness parameter (Shih et al., 1997; Tsang and Kong, 2001, p. 430). At last, the extinction and scattering coefficients are respectively given by:

$$\kappa_e = 2k_0\Im\sqrt{\epsilon_{\text{eff}}} \tag{61}$$

and

$$\kappa_s = \frac{2}{9}k_0^4 a^3 f_2 \left|\frac{\epsilon_2 - \epsilon_1}{1 + \frac{\epsilon_s - \epsilon_b}{3\epsilon_{\text{eff}}}(1 - f_2)}\right|^2 \frac{(1 - f_2)^4}{(1 + 2f_2 - tf_2(1 - f_2))^2} \tag{62}$$

where $\Im$ denotes the imaginary part of a complex number.

The effective permittivity in the QCA approximation is given by:

$$\epsilon_{\text{eff}} = \varepsilon_1 + 3\varepsilon_1 f_2 \frac{y}{1 - f_2 y}\left(1 + \jmath\frac{2}{3}(k_0 a)^3 \frac{y}{1 - f_2 y}\frac{(1 - f)^4}{(1 + 2f_2 - tf_2(1 - f_2))^2}\right)$$
$$y = \frac{\varepsilon_2 - \varepsilon_1}{\varepsilon_2 + 2\varepsilon_1} \tag{63}$$

and the extinction and scattering coefficients with:

$$\kappa_e = 2k_0\Im\sqrt{\epsilon_{\text{eff}}} \tag{64}$$

and

$$\kappa_s = \frac{2}{9}k_0^4 a^3 f_2 \left|\frac{\epsilon_{\text{eff}}}{e_1} - 1\right|^2 \frac{(1 - f)^4}{(1 + 2f_2 - tf_2(1 - f_2))^2} \tag{65}$$

In the short range approximation, the phase matrix of DMRT QCA and DMRT QCA-CP is the same as for independent Rayleigh scatterers.



*Author contributions.* The three authors have contributed to model development, validation and to writing the manuscript

*Acknowledgements.* We acknowledge the European Space Agency which supported this model development under ESTEC Contract No.4000112698/14/N
with a contribution from the NERC National Centre for Earth Observation. We would like to thank Christian Mätzler for numerous advices.





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





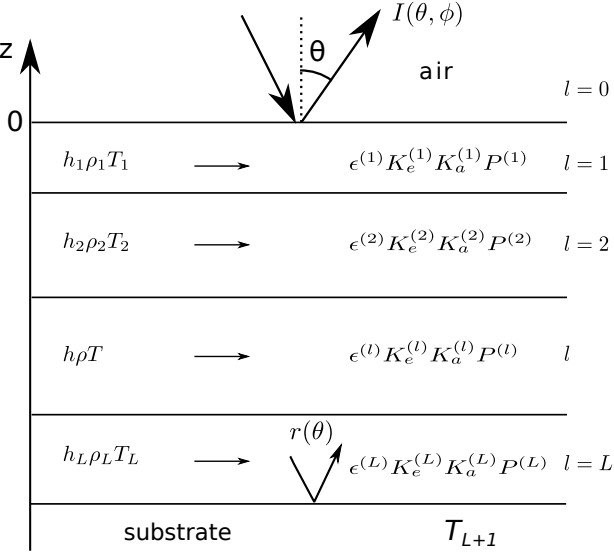

**Figure 1.** Multi-layered medium modeled by SMRT.

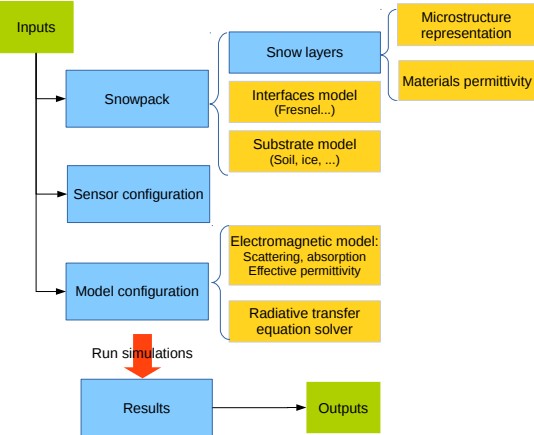

**Figure 2.** SMRT architecture and main components. The core components (blue) are fixed and contains no scientific code in contrast to the switchable and extensible components (orange) which define the snowpack and model configurations.



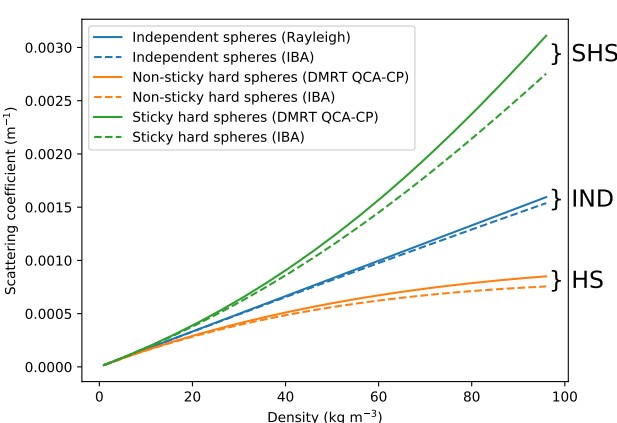

**Figure 3.** Scattering coefficient by several electromagnetic theories (independent spheres, IND; non-sticky hard spheres, HS; sticky hard spheres, SHS) as a function of density for sparse media described by various microstructure .





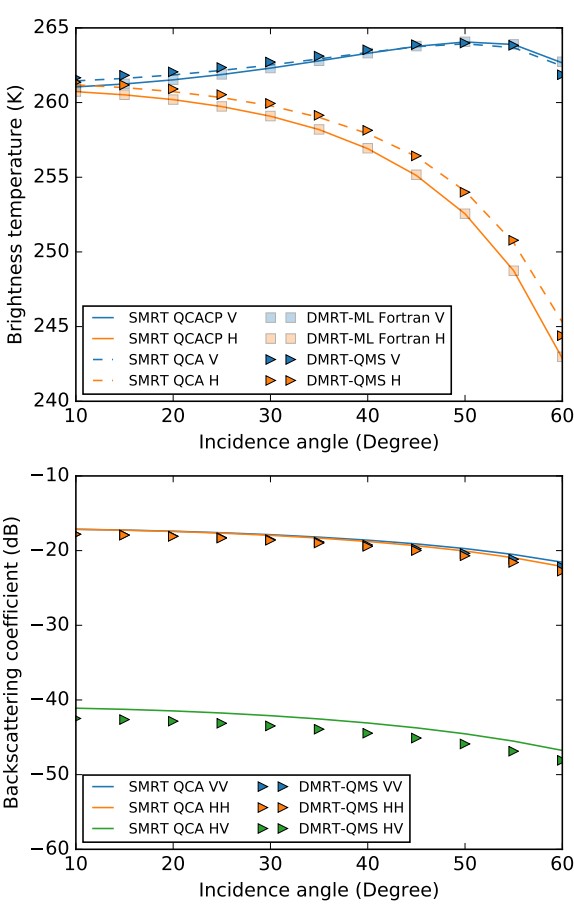

**Figure 4.** Comparison of brightness temperatures (top) and backscattering coefficients (bottom) simulated by SMRT QCA, SMRT QCA-CP, DMRT-ML and DMRT-QMS (when relevant for the active mode). The snowpack is semi-infinite with density of $300\,\mathrm{kg\,m^{-3}}$, sphere radius of $100\,\mu\mathrm{m}$, temperature of $265\,\mathrm{K}$ and stickiness of $0.5$. The frequency is $37\,\mathrm{GHz}$.





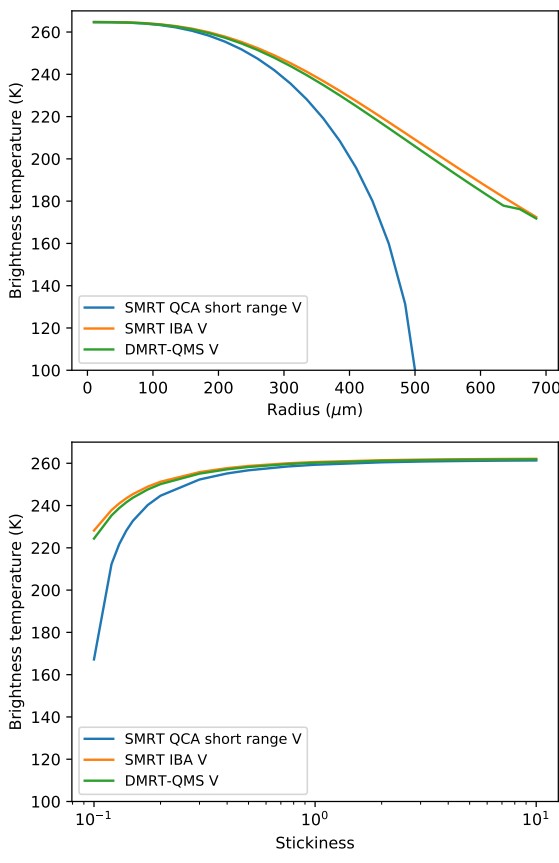

**Figure 5.** Brightness temperature at V polarization (55°) as a function of sphere radius and stickiness simulated by SMRT QCA, SMRT IBA with SHS microstructure and DMRT-QMS (QCA Mie) for the same snowpack as in Fig. 4.





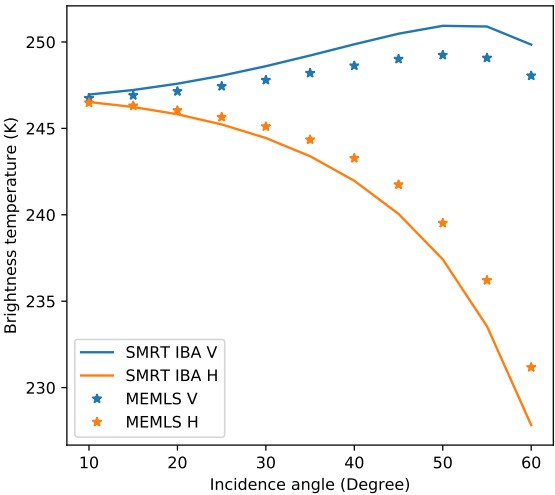

**Figure 6.** Comparison of brightness temperatures simulated by SMRT IBA with exponential autocorrelation function and MEMLS. The correlation length is $100\,\mu m$, other parameters are similar to those of Fig. 4.

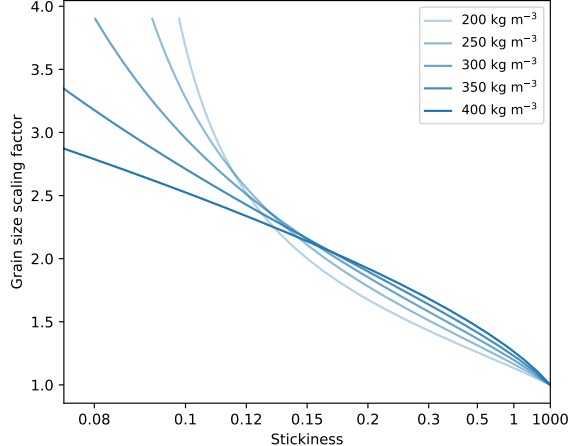

**Figure 7.** Grain size scaling factor $\phi_{\mathrm{SHS}}$ needed for simulations with non-sticky hard spheres with radius $\phi_{\mathrm{SHS}}a$ to yield the same brightness temperature at V polarization as simulations with sticky hard spheres with radius $a = 100\,\mu m$. Other parameters are similar to Fig. 4.





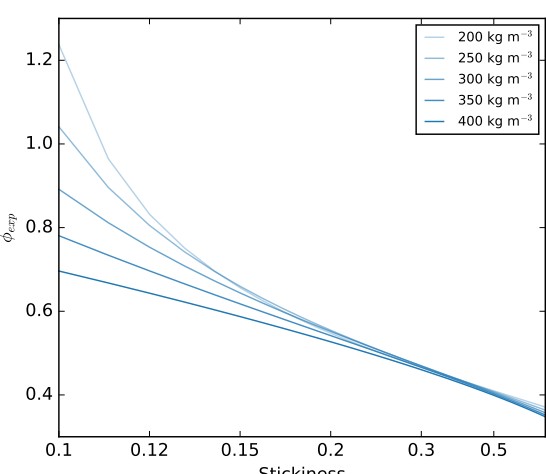

**Figure 8.** Scaling factor $\phi_{\mathrm{exp}}$ needed for simulations with exponential autocorrelation function with scaled correlation length $\phi_{\mathrm{exp}} \frac{4}{3} (1 - \rho/\rho_{ice}) a$ to yield the same brightness temperature at V polarization as simulations with sticky hard spheres with radius $a = 100\,\mu\mathrm{m}$. Other parameters are similar to Fig. 4.





**Table 1.** SMRT components and different formulations available in version 1.0

| Components | Formulations |
| --- | --- |
| Microstructure | Exponential |
| | Sticky Hard Spheres |
| | Independent Sphere |
| | Gaussian Random Field |
| | Teubner Strey |
| Substrate | Generic reflector |
| | Wegmüller Soil (Wegmüller and Mätzler, 1999) |
| | Flat surface |
| Interface between layers | Flat interface |
| | Fully transparent interface |
| Permittivity | Ice (Mätzler, 1987) |
| | Freshwater (Mätzler, 1987) |
| | Wet snow (Jin, 1994) |
| Electromagnetic model | Rayleigh |
| | DMRT QCA-CP short range |
| | DMRT QCA short range |
| | IBA |
| RT solver | DORT |
| Bindings to existing model | MEMLS (passive only) |
| | DMRT-QMS (passive only) |
| | HUT |