# Peer review of "SMRT: An active / passive microwave radiative transfer model for snow with multiple microstructure and scattering formulations (v1.0)"

_Geoscientific Model Development, 2017_

## Short Comment (SC1) · 21 Feb 2018

Comments on Geosci. Model Dev. Discuss., https://doi.org/10.5194/gmd-2017-314

SMRT: An active / passive microwave radiative transfer model for snow with multiple microstructure and scattering formulations (v1.0)

By Ghislain Picard, Melody Sandells, and Henning Löwe

Very useful tool for simulation of snow microwave emission, allowing interesting comparison between models. SMRT will certainly contribute to improve brightness temperature simulations, using measured and/or simulated snowpack characterization (stratification, micro-structure…).

I suggest to modify the Table 1, because, as said in the text (see p.11), all choices of microstructure parameters are not compatible with all choices of electromagnetic models!
I also suggest to add in this Table 1 the input parameters needed for running SMRT corresponding to each of the microstructure parametrization. The Fig.1 only gives the fundamental parameters used by the model.

For the IBA_exp mode, the definition of $l_{ex}$ is not clear (Eq. 17). In practice, as said in the text, in the field, the correlation length can only be estimated from easy measurable parameters, e.g. SSA and density!
The commonly used relationship is the Deby equation : $l_{ex}$ = 4 A
where A = $(1 - f) / (\rho_{ice}$ SSA)
In practice, previous studies and this SMRT paper show that, in general, a factor Φ must be used, such as:

$$l_{ex} = 4 \, \Phi \, A \quad (Eq.1)$$

Matzler et al used Φ = ¾, given : $l_{ex}$ = 4 ¾ A = 3 A

$$\text{or in general}: l_{ex} = 3 \, \Phi_{ex} \, A \quad (Eq.2)$$

I think that there could be a confusion here depending of the definition of the Autocorrelation function used (Eq.17) . Is it the same definition in MEMLS?
Montpetit et al. (2013) used Eq.1 for running MEMLS. The factor considered by Montpetit as input of MEMLS is not for Eq. 2 but for Eq. 1 ( Line 32, p16).
When applied to Eq2, this gives $\Phi_{ex}$ =1.3 x 4/3 = 1.73  instead of 0.975 as stated in the paper.  I suspect a mistake here?
Why Fig. 8 uses Eq.2 , instead of the original formula Eq. 1 ?  For clarity and to ovoid ambiguity, I suggest to plot the Fig. 8 using Eq.1 and not Eq.2. Text p17 should then be modified. (there is presently a typo error: 0.13 at 300 kg/m3, line 6)

I also suggest to better discuss or explain how to include an ice lens in the snowpack. This is a major issue because of the observed significant increase of winter heat wave events and of rain-on-snow events. Both events generate ice crust in the snowpack that have a strong impact on microwave emission.

Other comments:

- Defined the $\nu$ parameter (frequency) p5, line 31
- P9 Line 4 should need parenthesis? : $ka = ko\ f_2\ F(\varepsilon_2 Y^2)$
- P10 Eq. 17, 18, 19, 20 and 24 should be aligned?
- P27, Eq 65 : $e_1$ ?

Alain Royer
University of Sherbrooke, Québec, Canada

---

## Referee Comment (RC1) · Anonymous Referee #1 · 9 Mar 2018

The work described in the paper, and the model made freely available, represents an important effort for the community because it will be possible to analyze and compare the behavior of different e.m. models for the snow. The paper is well written with exaustive credit given to the authors of the original theories and methodologies. I have few minor observations:

- The paper claims that SMRT work for the active and passive case, however nothing is said about the former one. Just something in the introduction and in section 4 "Limitations..." I think the active case should be expanded as for the passive case

- Several papers have been published by Tsang about DMRT with a scatter size dis-

tribution. For instance https://doi.org/10.1163/156939392X01156 . In my opinion they should be included in the discussion for completeness

- Liang et al. 2008 deals with passive remote sensing, not active as stated on page 2.

- Table 1 should be better arranged showing which components can be freely chosen and which one must be used with a given formulation

- Diagram in figure 2 is not clear. It seems a mix between a functional description and a flowchart, however cannot be followed as a flowchart and neither it is clear the relationship between the blocks. It should be rearranged.

Provided these minor changes I think the paper is worth to be published.

―――――――――――――――

---

## Referee Comment (RC2) · C. Mätzler (Referee) · 12 Mar 2018

Comments to the manuscript of Picard, Sandells, and Löwe, 2018: SMRT

**General comment**:
This manuscript describes excellent work for the advancement of microwave remote sensing of snowpacks. The present version needs some corrections and improvements as described below. Furthermore I see a need for numerical validations, e.g. as proposed in Comment 15.
This review does not include the Appendices (except for p. 20).

**Special comments for the revision**:

Comment 1
The statement (p. 2, l. 10-11) on the influence of the atmosphere is not adequate because atmospheric effects can be quite significant and sometimes dominant. The authors can circumvent the problem by defining the boundary conditions at the snow surface. Instead of an illumination by constant cosmic background, the illumination also contains an atmospheric contribution, leading to a frequency-dependent sky brightness temperature $T_{sky}$. A main advantage of microwave radiation is that scattering in the atmosphere is negligible (except for precipitation). The introduction of Kirchhoff's Law on thermal emission, using emissivities and scene reflectivity (snow & substrate) together with an appropriate figure would improve the understanding. In addition the link with active radiation would become more apparent.

Comment 2
On p. 4, l. 27 it would be helpful to have references for Python and LGPLv3 License.

Comment 3
Equation (1) on p. 5 is the well-known radiative transfer equation for plane-parallel media (S. Chandrasekhar, Radiative Tranfer, 1950), here in the Rayleigh-Jeans Approximation. Unfortunately, in this form, it is only valid if the refractive index n=1. Since snow is a refractive medium with n>1, the equation needs modifications. For isotropic snow, the adaptation is simple. The specific intensity I has to be changed to its reduced value

$$I_1 = I/n^2, \qquad\qquad (C\ 1)$$

see e.g. the Fundamental Theorem of Radiometry, in Mobley, C.D., Light and Water (1994), or Hilbert, D., Die Begründung der elementaren Strahlungstheorie, Physik. Zeitschrift XII, 1056-1064 (1912). For anisotropic media, see e.g. Bekefi, G., Radiation Processes in Plasmas, New York, Wiley (1966). In a non-scattering and non-absorbing medium $I_1$ is a conserved quantity, but not I. Likewise, the source term $\alpha T(z)$ is to be divided by $n^2$ to get

$$\alpha_1 T(z), \text{ where } \alpha_1 = \alpha/n^2 = 2\nu k/c_0^2 \qquad\qquad (C\ 2)$$

Here $c_0$ is the speed of light in vacuum. Thanks to this correction, the emitting source term is a constant quantity in an isothermal environment, a requirement of thermodynamics. This is not true for $\alpha T(z)$ in a layered medium with n(z) changing with height.
It is possible that the authors made the necessary adaptation without being aware of, meaning that numerically, everything is OK. Still the formulation should be corrected. The adaptation is automatically taken into account in the formulation of temperatures (Rayleigh- Jeans) and brightness temperatures, instead of radiances.

The following page (extract from lecture notes) gives some more details.

*Slightly inhomogeneous medium*

Now we assume that the medium is slightly inhomogeneous, but scattering and absorption are absent (reflection and scattering are negligible if the gradient of the real part of the refractive index is sufficiently small: $|\nabla n'| \ll k$, and absorption is negligible if the imaginary part is $n''$=0). The rays are no longer straight lines, but follow the rules of geometric optics (Snell's Law, Fermat's Principle of the shortest path, Eikonal Equation). It can be shown (Hilbert, 1912; Mobley, 1994) that the following quantity is conserved:

$$I_{1\nu} = \frac{I_\nu}{n'^2}; \text{ thus } \frac{dI_{1\nu}}{ds} = 0; \text{ for the Stokes Vector } \mathbf{I}_{1\nu} = \frac{\mathbf{I}_\nu}{n'^2}; \frac{d\mathbf{I}_{1\nu}}{ds} = 0 \qquad (9.3)$$

Note that $n'$=$n$ because $n''$=0. For illustration and verification of (9.3), we investigate the situation of a one-dimensionally inhomogeneous medium where the refractive index deceases in a transition region with increasing height (Figure 9.2).

[Figure]

Figure 9.2: Power conservation for a refracted ray passing from one medium in another through dA. Reflection is avoided by a soft transition

$$\left|\frac{dn}{dx_3}\right| \ll \frac{1}{\lambda}$$

Power conservation requires $dP_1$=$dP_2$, thus

$$I_\nu(1,\theta_1,\phi_1)\cdot\cos\theta_1\cdot d\Omega_1\cdot dA\cdot d\nu = I_\nu(n_2,\theta_2,\phi_2)\cdot\cos\theta_2\cdot d\Omega_2\cdot dA\cdot d\nu \qquad (9.4)$$

From Snell's law we have $\sin\theta_1 = n_2\sin\theta_2$. Furthermore, since

$$d\Omega_1 = \sin\theta_1 d\theta_1 d\varphi, \quad d\Omega_2 = \sin\theta_2 d\theta_2 d\varphi, \quad \text{and} \quad \cos\theta_1 d\theta_1 = d(\sin\theta_1) = n_2 d(\sin\theta_2) = n_2\cos\theta_2 d\theta_2,$$

we get

$$\cos\theta_1 d\Omega_1 = n_2^2\cos\theta_2 d\Omega_2 \qquad (9.5)$$

Equations (9.4) and (9.5) lead to (9.3). Equation (9.3) also means that the Planck function is not conserved, but the following quantity is:

$$B_{1\nu} := \frac{B_\nu(\mathbf{r},T_b)}{(n'(\mathbf{r}))^2} = \frac{2h\nu^3}{c_0^2(\exp(h\nu/k_bT_b)-1)} = \text{constant} \qquad (9.6)$$

Since the quantities on the right side either are fundamental constants ($h$, $k_b$, $c_0$), a fixed frequency $\nu$, or the brightness temperature $T_b$, Eq. (9.6) means that $T_b$ does not change along the propagation path. Thus $I_{1\nu} = B_{1\nu}$ and $T_b$ are conserved quantities. This is a first important result, the *fundamental theorem of radiometry* (Mobley, 1994). If the brightness temperature $T_b$ did change, it would violate principles of thermodynamics.

Comment 4
Equation (4), last integral: integration interval must be changed to  µ' from 0 to +1.

Comment 5
In Equations (2) to (4) the variable µ appears as being the same in all layers. This is incorrect. The incidence angle (and thus µ) changes due to refraction from layer to layer. Refraction should be formulated explicitly and taken into account. Otherwise the connection fails at layer interfaces. Note that upon refraction the solid angle of beams is changing too.

Comment 6
p. 7, l. 16 -17: The depolarisation factors are defined with respect to the 3 main axes of the ellipsoid with $A_1+A_2+A_3 = 1$. Equation (6) gives the mean value of the squared-field ratio for an isotropic distribution of such ellipsoids. The situation with all $A_i=1/3$ corresponds to spherical scatterers.

Comment 7
p. 12, l. 4: after this illustration I expect a short description of what it means. Illustrative results are missing.

Comment 8
p. 12, l. 12: in addition, the temperature of the substrate is required (for the passive mode).

Comment 9
p. 13, l. 8-9: Improve sentence to „Different configurations can be explored by adapting the code provided as open source (see data availability)", and explain the missing part more clearly, using an additional sentence. Examples would help.

Comment 10
p. 14, l. 10: change „scattering coefficient" to „brightness temperature" (which is actually shown in Figure 5).

Comment 11
p. 17, l. 2-3: clarify „fixing density and SSA" in Figure 8. The caption to Figure 8 indicates a fixed radius of 0.1 mm.

Comment 12
p. 18, l. 25: delete „constructive" or add „and destructive" before „interferences" and add „for short phase differences"

Comment 13
p. 20, l. 11-12: what do you mean with „jupyter notebooks" ?
And explain the acronym „DORT"

Comment 14
p. 20, l. 20-22: The description of the treatment of streams in different layers is much too short to be understood here. It is related to my Comments 3 and 5 (above). Please improve this text and estimate the potential errors introduced by one or the other method.

Comment 15
Tests should be made to check how accurate the radiative-transfer code is. One simple check is by assuming an isothermal environment ($T_{sky} = T_{snow} = T_{substrate} = T$), and then computing internal brightness temperatures in all different directions and at different positions. If any of these results

differ from T, an error is indicated. Choose situations without, with weak and with strong volume scattering, and interface reflections, respectively.

Comment 16
Figure 5: add (print) the value of the stickiness parameter in the upper graph and the sphere radius in the lower graph.

**Typos**:
p. 5, l. 31: change „Planck constant" to „Boltzmann constant"
p. 6, l. 12 & 20: change ‚transmittivity' to ‚transmissivity' as used in the microwave range or ‚transmittance' as used in optics.
p. 6, l. 22: change „materials permittivity" to „material permittivity".
p. 7, l. 8: use symbol $c_0$ for speed of light in vacuum as proposed in (C 2).
p. 9, l. 1: change „will" to „with".
p. 13, l. 25: change 256 K to 265 K.
p. 14, l. 16: change „yields" to „yield"
p. 17, l. 23: delete „other" (written twice)
p. 19, l. 18: delete „have" before „highlighted"
p. 19, l. 24: change „numerically equivalence" to „numerical equivalence"
p. 20, l. 6: change „other representation" to „other representations"

---

## Short Comment (SC2) · 12 Mar 2018

The precise version of the code discussed in the manuscript must be made available. The current best practice is for this code to be uploaded to a public repository and a DOI assigned. The DOI should be cited in the manuscript. github is inadequate because it does not readily link to the precise version of the code. However, making github code citable is not difficult; see: https://guides.github.com/activities/citable-code/

---

## Author Comment (AC1) · 18 May 2018

**Response to Alain Royer**

I suggest to modify the Table 1, because, as said in the text (see p.11), all choices of microstructure parameters are not compatible with all choices of electromagnetic models! I also suggest to add in this Table 1 the input parameters needed for running SMRT corresponding to each of the microstructure parametrization. The Fig.1 only gives the fundamental parameters used by the model.

We have added the information about the compatibility of each microstructure representation with the electromagnetic models.

The input parameters of each microstructure with their definition is given in the equations in Section 2.3. This section is concise and clearly ordered. Adding the parameters in Table 1 would duplicate the information and would require to overload the table legend with the parameter definition which is non-trivial for many of them (all except radius and correlation length). We prefer to keep Table 1 self-consistent and concise.

For Table 1, we have added ellipses to the list of fundamental parameters to make clearer that the list of parameters is insufficient. It is impossible to show explicitly all the possible parametrisations offered by SMRT in this figure.

For the IBA_exp mode, the definition of lex is not clear (Eq. 17). In prac
tice, as said in the text, in the field, the correlation length can only be estimated from easy measurable parameters, e.g. SSA and density!
The commonly used relationship is the Deby equation :
lex= 4 A
where A = (1 –f) / (rho_ice SSA)
In practice, previous studies and this SMRT paper show that, in general, a factor phi must  be used, such as: lex= 4 phi A (Eq.1)
Matzler et al used phi = 0.75, given : lex = 4 3/4 A = 3 A
or in general: lex= 3 phi A (Eq.2)
I think that there could be a confusion here depending of the definition of the  Autocorrelation function used (Eq.17). Is it the same definition in MEMLS?

We have added a statement regarding the equivalence of Cex as employed here to MEMLSto comprehend the definition of lex. The same is used in MEMLS (in Matzler and Wiesman 1999, p 318 column a, in the text before equation 4).

Montpetit et al. (2013) used Eq.1 for running MEMLS. The factor considered by
Montpetit  as input of MEMLS is not for Eq. 2 but for Eq. 1 ( Line 32, p16). When applied to Eq2, this gives phi_ex =1.3 x 4/3 = 1.73 instead of 0.975 as stated in the paper. I suspect a mistake here?

We indeed have mis-interpreted Montpetit et al. 2013. We have corrected the text.

Why Fig. 8 uses Eq.2 , instead of the original formula Eq. 1 ? For clarity and to ovoid  ambiguity, I suggest to plot the Fig. 8 using Eq.1 and not Eq.2. Text p17 should then be modified. (there is presently a typo error: 0.13 at 300 kg/m3, line 6)

Fig 8 uses Debye formula scaled by phi which is your Eq.1  lex= 4 (1 –f) / (rho_ice SSA) but

expressed with the radius a=3/(rho_ice SSA), so that lex = 4/3 (1 –f)a.

The typo error is corrected.

I also suggest to better discuss or explain how to include an ice lens in the snowpack.
This is a major issue because of the observed significant increase of winter heat wave events and of rain-on-snow events. Both events generate ice crust in the snowpack that have a strong impact on microwave emission.

We have added a sentence referring to ice lenses in the IBA section: " This allows in particular the representation of pure ice lenses and ice crusts in the snowpack using IBA.". This is added in this section because MEMLS absorption coefficient is not compatible with high density. For this reason, we propose a different absorption formulation as a default (while still providing the original formulation for users interested by inter-comparison).

Other comments:
- Defined the nu parameter (frequency) p5, line 31

added

- P9 Line 4 should need parenthesis? : ka = ko f2F($\nearrow$2Y2)

The formula is correct with or without parenthesis as Y2 is a real number.

- P10 Eq. 17, 18, 19, 20 and 24 should be aligned?

The equations has been aligned.

- P27, Eq 65 : e1

Corrected.

---

## Author Comment (AC2) · 18 May 2018

**Response to Anonymous reviewer**

The work described in the paper, and the model made freely available, represents an important effort for the community because it will be possible to analyze and compare the behavior of different e.m. models for the snow. The paper is well written with exaustive credit given to the authors of the original theories and methodologies. I have few minor observations:

- The paper claims that SMRT work for the active and passive case, however nothing is said about the former one. Just something in the introduction and in section 4 "Limitations..." I think the active case should be expanded as for the passive case

The formulations in the paper are valid for both passive and active. In the results section, Fig 4 already included a computation for the active mode. We have added the calculation for Fig 5.

- Several papers have been published by Tsang about DMRT with a scatter size distribution. For instance https://doi.org/10.1163/156939392X01156. In my opinion they should be included in the discussion for completeness

We have included this reference in the section "On the equivalence of microstructure models" as follows:
"Though the approach of using a stickiness close to 0.1 seems more physical compared to an empirical scaling factor, it also has weaknesses. Natural snow is composed of grains with variable size, which more resembles a collection of spheres with a distribution of radii (i.e. poly-dispersed spheres). Such dispersion is important and generally leads to increased scattering compared to the medium with mono-disperse spheres having the mean radius of the poly-disperse spheres (Tsang and Kong, 1992). However, the analytical treatment of the ACF for poly-dispersed sticky hard spheres is tedious"

- Liang et al. 2008 deals with passive remote sensing, not active as stated on page 2.

It is removed

- Table 1 should be better arranged showing which components can be freely chosen and which one must be used with a given formulation

We have added the information in Table 1 in parenthesis for each microstructure. We have also added separator between the different components which makes the Table easier to read.

- Diagram in figure 2 is not clear. It seems a mix between a functional description and a flowchart, however cannot be followed as a flowchart and neither it is clear the relationship between the blocks. It should be rearranged.

We have removed the flowchart aspect and kept the components aspect.

Provided these minor changes I think the paper is worth to be published.

---

## Author Comment (AC3) · 18 May 2018

**Response to Christian Mätzler**

Comment 1
The statement (p. 2, l. 10-11) on the influence of the atmosphere is not adequate because atmospheric effects can be quite significant and sometimes dominant. The authors can circumvent the problem by defining the boundary conditions at the snow surface. Instead of an illumination by constant cosmic background, the illumination also contains an atmospheric contribution, leading to a frequency-dependent sky brightness temperature Tsky . A main advantage of microwave radiation is that scattering in the atmosphere is negligible (except for precipitation). The introduction of Kirchhoff's Law on thermal emission, using emissivities and scene reflectivity (snow & substrate) together with an appropriate figure would improve the understanding. In addition the link with active radiation would become more apparent.

The statement was moderated by "most frequencies" but for clarity reason it completely is removed. Most applications (which is the topic of this introductory paragraph) uses satellite data at frequencies (typically 19 and 37 GHz) where the atmosphere has a weak effect and is often neglected. The remaining of the comment seems to refer to Figure 1. We have added a symbol for the incoming beam and explained in the legend that it can be the radar beam or the atmospheric contribution. It is not clear why Kirchoff law should be introduced here. The snowpack is usually non-isothermal (as the atmosphere) which makes the notions of emissivities and Kirchoff law not fully adequate. This is the reason why we need to solve the radiative transfer equation with the thermal emission term, not just compute the reflectivity of the scene.

Comment 2
On p. 4, l. 27 it would be helpful to have references for Python and LGPLv3 License

We have added web link as the language and the license do not exist as citeable document.

Comment 3
Equation (1) on p. 5 is the well-known radiative transfer equation for plane-parallel media (S. Chandrasekhar, Radiative Tranfer, 1950), here in the Rayleigh-Jeans Approximation. Unfortunately, in this form, it is only valid if the refractive index n=1. Since snow is a refractive medium with n>1, the equation needs modifications. For isotropic snow, the adaptation is simple. The specific intensity I has to be changed to its reduced value $I_1 = I/n^2$, (C 1) see e.g. the Fundamental Theorem of Radiometry, in Mobley, C.D., Light and Water (1994), or
Hilbert, D., Die Begründung der elementaren Strahlungstheorie, Physik. Zeitschrift XII, 1056- 1064 (1912). For anisotropic media, see e.g. Bekefi, G., Radiation Processes in Plasmas, New York, Wiley (1966). In a non-scattering and non-absorbing medium $I_1$ is a conserved quantity, but not I. Likewise, the source term $\alpha T(z)$ is to be divided by $n^2$ to get $\alpha_1 T(z)$, where
$$\alpha_1 = \alpha/n^2 = 2vk/c_0^2 \quad (C\ 2)$$
Here $c_0$ is the speed of light in vacuum. Thanks to this correction, the emitting source term is a constant quantity in an isothermal environment, a requirement of thermodynamics. This is not true for $\alpha T(z)$ in a layered medium with n(z) changing with height.
It is possible that the authors made the necessary adaptation without being aware of, meaning that numerically, everything is OK. Still the formulation should be corrected. The adaptation is automatically taken into account in the formulation of temperatures (Rayleigh-Jeans) and brightness temperatures, instead of radiances. The following page (extract from lecture notes) gives some more details.

We have changed the text by renaming "specific intensity" to "reduced specific intensity" and given

the definition. The reference to Mobley 1994 has been added.
As stated in the text, the code is implemented with I=Tb and alpha=1 so that this mistake as no impact in the code.

Comment 4
Equation (4), last integral: integration interval must be changed to µ' from 0 to +1.

Yes, thank you to spot this mistake

Comment 5
In Equations (2) to (4) the variable µ appears as being the same in all layers. This is incorrect. The incidence angle (and thus µ) changes due to refraction from layer to layer. Refraction should be formulated explicitely and taken into account. Otherwise the connection fails at layer interfaces. Note that upon refraction the solid angle of beams is changing too

We have made the refraction explicit by adding a function to convert the angles between layers and introducing the S function (for details cf manuscript) to apply the Snell-Descartes law.

Comment 6
p. 7, l. 16-17: The depolarisation factors are defined with respect to the 3 main axes of the ellipsoid with A1+A2+A3 = 1. Equation (6) gives the mean value of the squared-field ratio for an isotropic distribution of such ellipsoids. The situation with all Ai=1/3 corresponds to spherical scatterers.

We have changed the sentence "In SMRT version 1.0, only isotropic microstructures are considered which implies Aj= 1/3."
into
"In SMRT version 1.0, only spherical scatterers are considered which implies Aj= 1/3."

Comment 7
p. 12, l. 4: after
this illustration I expect a short description of what it means. Illustrative results are missing

We have added a description and the numerical result.

Comment 8 p. 12, l. 12: in addition, the temperature of the substrate is required (for the passive mode).

This information is added

Comment 9 p. 13, l. 8-9: Improve sentence to „Different configurations can be explored by adapting the code provided as open source (see data availability)", and explain the missing part more clearly, using an additional sentence. Examples would help.

We have reformulated the sentence. The provided codes are well documented, changing the frequency or the snowpack properties is straightforward.

Comment 10 p. 14, l. 10: change „scattering coefficient" to „brightness temperature" (which is actually shown in Figure 5).

corrected.

Comment 11 p. 17, l. 2-3: clarify „fixing density and SSA" in Figure 8. The caption to Figure 8 indicates a fixed radius of 0.1 mm.

The sentence is rephrased: "Each curve is obtained, for a given density, by optimizing phi_exp to obtain equivalence between"

Comment 12 p. 18, l. 25: delete „constructive" or add „and destructive" before „interferences" and add „for short  phase differences"

corrected as suggested.

Comment 13 p. 20, l. 11-12:  what do you mean with „jupyter notebooks" ? And explain the acronym „DORT"

we have reformulated and added a reference for jupyter notebooks. DORT in the Annex title has been expanded.

Comment 14 p. 20, l. 20-22: The description of the treatment of streams in different layers is much too short to be understood he re. It is related to my Comments 3 and 5 (above). Please improve this text and estimate the potential errors introduced by one or the other method.

We have added a precise description in  the Appendix (page 22) using the newly introduced function S (your comment 5)

Comment 15
Tests should be made to check how accurate the radiative-transfer code is. One simple check is by assuming an isothermal environment (Tsky= Tsnow= Tsubstrate = T), and then computing internal brightness temperatures in all different directions and at different positions. If any of these results differ from T, an error is indicated. Choose situations without, with weak and with strong volume scattering, and interface reflections, respectively.

Internal numerical tests in SMRT has been added to check this situation of isothermal model
https://github.com/smrt-model/smrt/blob/master/smrt/test/test_physics_law.py.
We have also added a test on Kirchoff law (1 - reflectivity =emissivity) for opaque and isothermal surface. These tests are run for high and low scattering media and for thick and shallow snowpacks.

The results are accurate within a 0.01K tolerance for the isothermal situation and 0.002 in emissivity for the kirchoff law.

---

## Author Comment (AC4) · 18 May 2018

Dear Editor,

We already had obtained a DOI for the discussion paper version: DOI: 10.5281/zenodo.1173104 The latest release which is relevant for the final paper is: DOI: 10.5281/zenodo.1249413. This information is added in the paper.

---

## Author Response (AR2)

We have implemented the latest comments from Christian Mätzler as follows:

1) On p. 5 of the pdf version, line 26, in the definition of the reduced specific intensity: change "I = I'/n" to "I = I'/n^2" , (i.e. replace n by n-square).

done

2) On p. 6, first line: change „Planck" to „Boltzmann" .
Note that the Planck constant does not appear in the classical Rayleigh-Jeans formula!

done

3) On p. 7, line 4: change „materials permittivity formulations" to „material-permittivity formulations"
Note that adjectives do not have a plural form in English.

done

4) Figure 5 was extended to 4 square plots. This is fine. However the plots are too small and should be increased to a size that allows proper reading of everything.

We have increased the size in the pdf, but we expect the size adjustment will be done by the editor while preparing the proof. We will carefully check the quality of the graph when reviewing the proof. The figure is in a vectorial format and will scale without any degradation.